CERN-TH-2024-007

PREPARED FOR SUBMISSION TO JHEP

# BPS Wilson loops in mass-deformed ABJM theory: Fermi gas expansions and new defect CFT data

**Elisabetta Armanini,**[a,b] **Luca Griguolo,**[c] **Luigi Guerrini**[d,e]

[a]*Theoretical Physics Department, CERN, 1211 Geneva 23, Switzerland*

[b]*Fields and Strings Laboratory, Institute of Physics, École Polytechnique Fédéral de Lausanne (EPFL), Route de la Sorge, CH-1015 Lausanne, Switzerland*

[c]*Dipartimento SMFI, Università di Parma and INFN Gruppo Collegato di Parma, Viale G.P. Usberti 7/A, 43100 Parma, Italy*

[d]*Dipartimento di Fisica, Università di Firenze and INFN Sezione di Firenze, via G. Sansone 1, 50019 Sesto Fiorentino, Italy*

[e]*Faculty of Physics, University of Warsaw, ul. Pasteura 5, 02-093 Warsaw, Poland*

*E-mail:* elisabetta.armanini@cern.ch, luca.griguolo@unipr.it, Luigi.Guerrini@fuw.edu.pl

ABSTRACT: We compute the expectation values of BPS Wilson loops in the mass-deformed ABJM theory using the Fermi gas technique. We obtain explicit results in terms of Airy functions, effectively resumming the full 1/N expansion up to exponentially small terms. In the maximal supersymmetric case, these expressions enable us to derive multi-point correlation functions for topological operators belonging to the stress tensor multiplet, in the presence of a 1/2–BPS Wilson line. From the one-point correlator, we recover the ABJM Bremsstrahlung function, confirming nicely previous results obtained through latitude Wilson loops. Likewise, higher point correlators can be used to extract iteratively new defect CFT data for higher dimensional topological operators. We present a detailed example of the dimension-two operator appearing in the OPE of two stress tensor multiplets.

# 1 Introduction

The maximally superconformal gauge theory in three dimensional space-time, commonly known as ABJM theory [1, 2], is an important theoretical laboratory in quantum field theory. It stands out in particular due to its tight relations with string and M-theory [3–7], a hidden integrability at large $N$ [8–10], as well as its relevance within the bootstrap program [11–14]. Moreover supersymmetric localization provides a large number of exactly computable quantum observables for ABJM [15], as Wilson loops [16–20] and the Bremsstrahlung function [21–25]. Compared with its four-dimensional cousin, $\mathcal{N} = 4$ SYM, it appears somehow less constrained by the manifest superconformal as well as hidden symmetries, making its study harder for specific observables as scattering amplitudes [26–28]. Also the space of its solvable massive deformations is larger [29, 30], resulting in a more intricate holographic description in this case [31, 32].

A peculiar property of ABJM theory is that it admits an interpretation, even in the presence of mass deformations, as a statistical system of $N$ one-dimensional non-interacting fermions [33]. Localization reduces the ABJM partition function on $S^3$ to a complicated two-matrix model [15]: the planar free energy and the subleading $1/N$ corrections in the standard 't Hooft expansion were determined in a pioneering series of papers long ago [34, 35]. The related genus-weighted series makes contact with the dual gravitational interpretation: it encodes the type IIA reduction of M-theory on the background $\mathrm{AdS}_4 \times \mathbb{CP}^3$ and it captures all worldsheet instanton corrections to the partition function [36]. However, in order to make contact with the M-theory regime, one has to study the ABJM matrix model in the so-called M-theory limit, where $N$ is large but the coupling constant $k$ is fixed. This was first attempted in [37], where the leading, large $N$ limit was studied. In order to understand in more detail the M-theory expansion, and the corrections to the large $N$ limit, the Fermi gas formulation has been instrumental [38–40]. In this approach, the Planck constant of the quantum gas is naturally identified with the inverse string coupling, and the semiclassical limit of the gas corresponds then to the strong string coupling limit in type IIA theory. One of the main virtues of the Fermi gas approach is that it makes it possible to calculate systematically non-perturbative stringy effects [41], opening the way for a quantitative determination of these contributions in the M-theory dual to ABJM theory. One can also compute exactly the vacuum expectation value (vev) of 1/6 and 1/2–BPS Wilson loops in ABJM theory using Fermi gas technique: these studies were initiated in [42] and further developed in [43] particularly for the 1/2–BPS Wilson loops. Non-perturbative instanton corrections were instead carefully examined in [44], highlighting some peculiar differences between the two cases. Interesting advances concerning the structure of large $N$ ABJM theory and its BPS Wilson loops, in different regimes, have been recently obtained both from the holographic [45] and the Fermi gas [46] perspectives.

The mass-deformed case was instead examined from the Fermi gas point of view in [47], where the partition function with two-deformation parameters has been obtained. The large $N$ behaviour of this partition function received a certain amount of attention in the last few years from different point of views: phase structure [48, 49], non-perturbative contributions [50] and dual string theoretical description [51]. Quite surprisingly Wilson

loops have never been studied within this approach and also their structure at holographic level remains largely unexplored: we plan to fill this gap in the present paper. Actually our motivation is two-fold. On the one hand, we aim to derive explicit large $N$ expansions for BPS Wilson loops, both in the t' Hooft and M-theory limit, providing precision data for holographic investigations. On the other, we would like to take advantage of the possibility to perform derivatives with respect to the deformation parameters to compute correlation functions of particular local operators in the presence of a 1/2–BPS Wilson line in ABJM theory. This approach has been already applied in the absence of the Wilson line [12, 52] and it provided a full set of OPE data in different situations [53]. Our proposal here relies heavily on the recent construction of a BPS configuration involving both the 1/2–BPS Wilson lines and a class of topological operators of dimension one, living on a linked circle [54]. Thanks to the supersymmetric Ward identity obtained in [55, 56], the 1/2–BPS Wilson loops in the massive deformed theory can be used as a functional generator for these correlators. We derive here the large $N$ expression for the one-point and the two-point functions and, as an application, we use them to recover a new one-point function of a dimension-two topological operator in the background of the Wilson line.

The structure of the paper is the following. In Section 2 we recall some basic facts about the massive deformations of ABJM theory and its BPS Wilson loops, setting up our notation. We start our computations in Section 3, exploring the large $N$ limit of the BPS Wilson loops directly from the relevant saddle-point equations for the matrix model. We obtain the leading order exponential behaviour as an independent result that should be reproduced in the Fermi gas approach. A review of the Fermi gas formalism is given in Section 4. Section 5 is instead devoted to the technical computation of 1/6 and 1/2 supersymmetric Wilson loops in the two-parameter deformed model, using the Fermi gas technique. In Section 6 we perform the explicit 't Hooft-genus expansion and the M-theory limit, recovering the known results in the undeformed situation. The applications of our results to correlation functions in the background of the Wilson line are contained in Section 7. We obtain an exact large $N$ expression for the one-point function of dimension one topological operators, that reproduces the Bremsstrahlung function in this regime [24]. From applying OPE to the two-point function we derive a new one-point datum, that we hope will be useful in future applications of bootstrap to the defect CFT defined arising in the presence of the Wilson line. Our conclusions and the possible extensions of the present work are discussed in Section 8. A certain number of Appendices, devoted to various technical aspects of our computations, complete the paper.

## 2 Mass-deformed ABJM theory and Wilson loops: generalities

In this section, we introduce the ABJM model, focusing on the exact results coming from localization. We recall localization results for supersymmetric real mass deformations and BPS Wilson loops and set the stage for the forthcoming computations.

## 2.1 The mass-deformed theory

The ABJM model is an $\mathcal{N} = 6$ Chern-Simons matter theory with gauge group $U(N) \times U(N)$. For each gauge group factor, there is a Chern-Simons kinetic term with integer level $k$ and $-k$, respectively[1]. Thus, the field content includes two gauge fields, $A_\mu$ and $\hat{A}_\mu$, transforming in the adjoint of the first and the second $U(N)$ factor of the gauge group. They are minimally coupled to four matter multiplets $C_I$, $\bar{\psi}^I$, $I = 1, \ldots, 4$ transforming in the bifundamental of the gauge group and their conjugates $\bar{C}^I$, $\psi_I$ transforming in the antibifundamental. Matter interactions are encoded into a suitable scalar potential and related Yukawa terms.

The theory possesses an $SU(4)_R \times U(1)_b$ R-symmetry. The $SU(4)_R$ is the R-symmetry group and acts on the fundamental indices $I = 1, \ldots, 4$, while $U(1)_b$ is the hidden topological symmetry generated by the diagonal ungauged $U(1)$ group factor. The latter will not play any role in our paper and we can safely ignore it for the sake of simplicity.

To introduce the supersymmetric deformations and the localization procedure it is useful to think of ABJM as a special $\mathcal{N} = 2$ theory. As showed in [4], it is possible to write the action in an $\mathcal{N} = 2$ language: the gauge content is given by two vector multiplets $\mathcal{V} = (A_\mu, \sigma, \lambda, \bar{\lambda}, D)$, while the matter is represented by two bifundamental chirals $A_i$ and two anti-bifundamental chirals $B_i$. In that formulation, only the Lagrangian $U(1)_\ell \times SU(2) \times SU(2)$ symmetry is visible. The $U(1)_\ell$ factor is the $\mathcal{N} = 2$ explicit R-symmetry, the $SU(2) \times SU(2)$ appearing as a flavor symmetry rotating the scalars $A_i$ and $B_i$ separately. Of course, integrating out the auxiliary fields we recover the R-symmetry of $\mathcal{N} = 6$ theory in its completeness. The conformal symmetry is preserved at the quantum level, thus promoting the theory to be fully superconformal.

A deformation which breaks down the conformal symmetry preserving instead SUSY consists in assigning different $U(1)_\ell$ R-charges $\Delta_{A_i}$, $\Delta_{B_i}$ to the chiral fields. The new R-charges are constrained by the structure of the superpotential $W \sim \epsilon^{ijkl} \text{Tr}(A_i A_j B_k B_l)$, which must have R-charge 2, that is

$$\Delta_{A_1} + \Delta_{A_2} + \Delta_{B_1} + \Delta_{B_2} = 2 \,. \tag{2.1}$$

In general, the space of these deformations is three-dimensional: in this paper we will discuss a particular subclass of the possible deformed models, i.e. ones described by two-parameters only. In this case, a convenient parametrization of the R-charges is the following

$$\Delta_{A_1} = \frac{1 + \zeta_2}{2} > \Delta_{A_2} = \frac{1 - \zeta_1}{2} \,, \qquad \Delta_{B_1} = \frac{1 + \zeta_1}{2} > \Delta_{B_2} = \frac{1 - \zeta_2}{2} \,, \tag{2.2}$$

with $-1 < \zeta_{1,2} < 1$. We can think of this non-canonical R-charge assignment as a redefinition of the $U(1)_\ell$ charge by mixing it with the Cartan subalgebra of the flavor symmetry.

In order to perform supersymmetric localization, we have to define the theory with this new R-symmetry on the 3d sphere $S^3$ [57]. The resulting model will be invariant under $SU(2|1)_L \times SU(2)_R$, whose bosonic part contains the isometry group $SO(4) \simeq SU(2)_L \times SU(2)_R$ of $S^3$ and our R-charge $U(1)_\ell$. Alternatively, one can think of the

---

[1]When $k = 1, 2$ the SUSY is enhanced to $\mathcal{N} = 8$. We will not discuss that in the paper.

R-charge deformations as a real mass deformation. The latter arises by coupling the theory to a background flavor vector multiplet for the symmetries mixing with the original R-symmetry and taking the SUSY-preserving rigid limit in which we set to zero all the (flavor) gauginos and their variations. The scalars in the multiplet get the only non-zero vev, which corresponds to a real mass deformation in the Lagrangian. It turns out that an imaginary value for the masses coincides with R-symmetry deformations of the type discussed above.

$\mathcal{N} = 2$ supersymmetric theories on $S^3$ are generally amenable to localization. In particular it is possible to reduce the partition function and, as we will discuss in the next subsection, the expectation values of some protected observables to finite dimensional integrals, i.e. matrix models. For the ABJM model, with our R-charge assignment the result is [15, 57, 58]

$$Z = \int \left( \prod_{i,j} \frac{d\lambda_i}{2\pi} \frac{d\mu_j}{2\pi} \right) e^{\frac{ik}{4\pi} \sum_i (\lambda_i^2 - \mu_i^2)} \frac{\prod_{i<j} 2\sinh(\lambda_i - \lambda_j) 2\sinh(\mu_i - \mu_j)}{\prod_{i,j} 2\cosh\left(\frac{\lambda_i - \mu_j - i\pi\zeta_1}{2}\right) 2\cosh\left(\frac{\lambda_i - \mu_j + i\pi\zeta_2}{2}\right)}, \quad (2.3)$$

reducing smoothly to the undeformed case as $\zeta_1, \zeta_2 \to 0$. In our case, we stress that the matrix model is still expressed in terms of elementary functions, which does not generally hold for deformations with three parameters.

## 2.2 Wilson loops

In supersymmetric models, BPS Wilson loops are a natural family of extended operators with the attractive perspective of being exactly computable. To make the standard Wilson loop supersymmetric, one needs to twist its connection with suitable combinations of additional fields [59]. Applying that logic to ABJM, we can define the so-called bosonic Wilson loop, whose expression is

$$W_R^{1/6}[\gamma] = \mathrm{Tr}_R \left[ \mathcal{P} \exp \left[ -i \oint_\gamma d\tau \left( A_\mu \dot{x}^\mu - \frac{2\pi i}{k} |\dot{x}| \mathcal{M}_I{}^J C_J \overline{C}^I \right) \right] \right] . \quad (2.4)$$

where $\gamma$ is a generic curve and $\mathcal{M}_I{}^J$ is an arbitrary matrix. To apply localization, we consider the maximally supersymmetric case, that is, $\gamma$ is a circle or a line, and $\mathcal{M}_I{}^J = \mathrm{diag}(1, -1, 1, -1)$. It is not hard to see that one can define a Wilson loop with the same property for the other $U(N)$ factor of the gauge group by starting with the corresponding gauge field, that is

$$\hat{W}_R^{1/6} = \mathrm{Tr}_R \left[ \mathcal{P} \exp \left[ -i \oint d\tau \left( \hat{A}_\mu \dot{x}^\mu - \frac{2\pi i}{k} |\dot{x}| \mathcal{M}_I{}^J \overline{C}^I C_J \right) \right] \right] . \quad (2.5)$$

In the massless case, both operators preserve four supercharges and are 1/6–BPS [16, 17, 60, 61]. The mass deformations partially change the situation: superconformal supercharges are explicitly broken, and the loop preserves two supercharges.

A peculiar property of supersymmetric Chern-Simons matter theories, including ABJM, is the possibility of building Wilson loops preserving more supersymmetry by incorporating matter fermions in the Wilson loop connection [18]. The resulting operator, known as the

fermionic Wilson loop, can be written as the superholonomy of the supergroup $U(N|N)$, which contains the original gauge group $U(N) \times U(N)$ as its diagonal part. Its explicit expression is

$$W_R[\gamma] = \mathrm{STr}_R \, \mathcal{P} \exp\left(-i \oint_\gamma d\tau \, L(\tau)\right), \tag{2.6}$$

where, for the moment, we are loose in the definition of the STr. Demanding local supersymmetry, one can fix the general structure for the superconnection [62]

$$L(\tau) = \begin{pmatrix} \mathcal{A} & i\sqrt{\frac{2\pi}{k}}|\dot{x}|\eta_I \overline{\psi}^I \\ -i\sqrt{\frac{2\pi}{k}}|\dot{x}|\psi_I \overline{\eta}^I & \hat{\mathcal{A}} \end{pmatrix}, \tag{2.7}$$

with

$$\begin{cases} \mathcal{A} = A_\mu \dot{x}^\mu - \frac{2\pi i}{k}|\dot{x}|M_J{}^I C_I \overline{C}^J \\ \hat{\mathcal{A}} = \hat{A}_\mu \dot{x}^\mu - \frac{2\pi i}{k}|\dot{x}|M_J{}^I \overline{C}^J C_I \end{cases}. \tag{2.8}$$

The amount of supersymmetry depends on the explicit choices of the path $\gamma$ and the various couplings, namely $M_I{}^J$, $\eta_I^\alpha$, and $\bar{\eta}_\alpha^I$.

Choosing the curve to be a circle or a line, we obtain a $\frac{1}{2}$–BPS operator, which is the most appealing case in view of the applications in holography and defect CFTs of Sections 6 and 7. Let us focus on a Wilson loop supported along a circle on the $x_2$, $x_3$ plane, parametrized by an angular coordinate $\tau$. If we take $M_J{}^I = \mathrm{diag}(-1, 1, 1, 1)$ and [2]

$$\eta_I^\alpha = n_I \eta^\alpha = \frac{e^{i\tau/2}}{\sqrt{2}} \begin{pmatrix} \sqrt{2} \\ 0 \\ 0 \\ 0 \end{pmatrix}_I (1, -ie^{-i\tau})^\alpha, \tag{2.9}$$

$$\overline{\eta}_\alpha^I = \overline{n}^I \overline{\eta}_\alpha = i(\eta_I^\alpha)^\dagger,$$

the loop preserves half of the supersymmetry. Similarly, one can define a $\frac{1}{2}$–BPS Wilson line, for instance, along the $x_3$ line. A possible way to determine the couplings is to perform a conformal transformation that maps the circle to a straight line. The scalar couplings $M_I{}^J$ are left unchanged, unlike the fermionic ones, which become constant vectors

$$\eta_I^\alpha = \sqrt{2}\delta_I^1 \delta_1^\alpha, \qquad \overline{\eta}_\alpha = i(\eta_I^\alpha)^\dagger. \tag{2.10}$$

A final subtlety has to be fixed to ensure supersymmetry: since we impose the weaker condition that the supersymmetric variation of the connection is a total derivative [62], the BPS equations are sensible to the periodicity of the fermions along the path. However, we can compensate for the lack of periodicity by multiplying the superholonomy by the

---

[2]More recently, the fermion couplings were interpreted as a family of defect marginal deformations, depending on continuous parameters. Here, we tuned them to have a maximal BPS operator [19, 63–66].

so-called twist matrix $T = \text{diag}(1, -1)$. That gives the correct prescription for the trace in (2.6). Likewise, we adopt the same prescription for the Wilson line. As in the bosonic case, mass deformations break some bulk supersymmetry, but the fermionic operators still preserve half of the remaining supercharges.

One of the features that make BPS Wilson loops an ideal playground to study non-perturbative aspects of QFT is localization [15], even in the presence of masses. Localization associates to the vev of bosonic BPS Wilson loop an operator insertion into the ABJM matrix model of (2.3). If we take $\Lambda$ to be the diagonal matrix $\Lambda = \text{diag}(\lambda_1, \ldots, \lambda_N)$, the insertion for (2.4) is the character

$$\langle W_R^{1/6} \rangle = \text{Tr}_R \, e^\Lambda \,. \tag{2.11}$$

That holds regardless of the presence of BPS mass deformations, which affects only the form of the matrix model, but not the insertion. When fermions are present, one can exploit a cohomological argument [18] and relate the vev of fermionic loops to those of purely bosonic ones thanks to

$$W_R^{1/2} = (W_R^{1/6} + \hat{W}_R^{1/6}) + \delta V \,, \tag{2.12}$$

for a given functional $V$ and a supercharge $\delta$ shared among the bosonic and fermionic operators [3]. That argument was later refined and made more transparent in [67]. In particular, it ensures its validity for any known supersymmetric bulk deformations, including the masses. Therefore, the expectation value of the fermionic loop corresponds to the following matrix average

$$\langle W_R^{1/2} \rangle = \langle W_R^{1/6} \rangle - (-1)^n \langle \hat{W}_R^{1/6} \rangle \,, \tag{2.13}$$

where $n$ is the winding number of the Wilson loop. For the sake of simplicity, and having in mind holographic applications, from now on we shall focus on the fundamental representation, for which the single-winding Wilson loop insertion is

$$\text{Tr}_\square \, e^\Lambda = \sum_{i=1}^N e^{\lambda_i} \,. \tag{2.14}$$

In the next section, we initiate a systematic analysis of those insertions.

## 3 The large $N$ limit from the eigenvalue density

In this section, we begin to explore BPS Wilson loops in the presence of mass deformations, at strong coupling. As it will be clear later, solving the system is not only an example of a computation beyond perturbation theory, but it provides also a distinguished set-up of non-conformal holography. From the mass deformation, we can further gain information on the defect conformal theories defined by Wilson loops in ABJM theory [54].

---

[3]The explicit forms of $V$ and $\delta$ are not relevant to this paper but can be found in [18].

Our starting point is the matrix model description (2.3), which is solvable when $N$ becomes large and $k$ is fixed. In this limit, one can assume that the discrete spectrum of eigenvalues $\lambda_i$, $i = 1, \ldots, N$, condensates and it is characterized by a continuous distribution $\rho(x)$ [68]. Thus, we can simply approximate any operator insertion $f(\lambda_i)$, with an integral

$$\langle \sum_i f(\lambda_i) \rangle_{\mathrm{MM}} = \int_C dx \, f(x) \rho(x) \, . \tag{3.1}$$

where we presumed $\rho$ to have a compact support $C$. The advantage of this approach stands in its simplicity: it is often possible to derive an explicit form for $\rho(x)$ by looking for saddle points at large $N$, namely by finding the distribution that extremizes the free energy $F$.

In this section, we apply this procedure to our matrix model. We shall consider the leading order behaviour when $N \to \infty$ while $k$ is kept fixed. As we will discuss later, this regime is connected to the so-called M-theory limit. Even if it does not correspond to the usual planar limit of the matrix model, the eigenvalue distribution was determined even in the massive case [4]. In the following, we will briefly recall it and compute the Wilson loop in this strict large $N$ limit.

## 3.1 The set-up

We leverage on the method developed in [37, 69], a technique that found its principal application in the computation of the free energy $F(\Delta) = -\log Z(\Delta)$, where $\Delta$ is a collective notation for some deformation parameters. Here, we will use it to extract the leading behavior of the 1/6–BPS Wilson loop in the presence of mass deformations.

To begin with, we review the derivation of the eigenvalue distribution of [69]. One of the crucial points is to assume the following behaviour for the eigenvalues at large $N$

$$\lambda_i = \sqrt{N} x_i + i y_i^{(1)} \, , \qquad \mu_i = \sqrt{N} x_i + i y_i^{(2)} \, , \tag{3.2}$$

a property that has been confirmed by the precise numerical analysis of [37]. Under the usual hypothesis that the eigenvalues become dense, we approximate them with the continuous functions $x(s), y^{(1,2)}(s) : [0, 1] \to \mathbb{R}$, where $s \in [0, 1]$ is a continuous parameter corresponding to $i/N$. We also slightly modify the standard notation and identify $\rho \equiv \frac{\mathrm{d}s}{\mathrm{d}x}$ with the density of the real part of the eigenvalues.

Implementing those assumptions into the matrix model [37, 69], one can express the free energy as a functional of $\rho(x)$ and $\delta y(x) = y^{(1)}(x) - y^{(2)}(x)$, namely

$$F[\rho, \delta y] = N^{\frac{3}{2}} \left[ \frac{k}{2\pi} \int dx \, x \rho(x) \delta y(x) - \int dx \, \rho(x)^2 \Big[ \delta y(x)^2 + \pi \delta y(x) \left( \zeta_1 - \zeta_2 \right) \right.$$
$$\left. + \frac{1}{2} \pi^2 \left( \zeta_1^2 + \zeta_2^2 - 2 \right) \Big] - \frac{\alpha}{2\pi} \left( \int dx \, \rho(x) - 1 \right) \right] \, , \tag{3.3}$$

where $\alpha$ is a Lagrange multiplier imposing the normalization of the distribution $\rho$. Notice that the expression of $F$ is local in $x$, a remarkable simplification due to the strict large $N$ limit.

---

[4] See [35] for a complete analysis of the planar limit in the undeformed case.

Solving the extremization problem for $F$ w.r.t. $\rho$ and $\delta y$, under the condition $\rho(x) > 0$, one finds two continuous piecewise smooth function

$$-\frac{\alpha}{\pi k \left(1 - \zeta_1\right)} < x < -\frac{\alpha}{\pi k \left(1 + \zeta_2\right)} \ :$$

$$\rho = \frac{\alpha + \pi k(1 - \zeta_1)x}{2\pi^3 \left(2 - \zeta_1 - \zeta_2\right)\left(\zeta_1 + \zeta_2\right)} \,, \qquad \delta y = \pi(\zeta_1 - 1) \,, \qquad (3.4)$$

$$-\frac{\alpha}{\pi k \left(1 + \zeta_2\right)} < x < \frac{\alpha}{\pi k \left(1 + \zeta_1\right)} \ :$$

$$\rho = \frac{\alpha + \frac{\pi}{2} xk \left(\zeta_2 - \zeta_1\right)}{\pi^3 \left(2 - \zeta_1 - \zeta_2\right)\left(\zeta_1 + \zeta_2 + 2\right)} \,, \qquad (3.5)$$

$$\delta y = \frac{k\pi^2 x \left(2 - \zeta_1^2 - \zeta_2^2\right) - \alpha\pi(\zeta_2 - \zeta_1)}{2\alpha - \pi k x(\zeta_1 - \zeta_2)} \,,$$

$$\frac{\alpha}{\pi k \left(1 + \zeta_1\right)} < x < \frac{\alpha}{\pi k \left(1 - \zeta_2\right)} \ :$$

$$\rho = \frac{\alpha - \pi k x(1 - \zeta_2)}{2\pi^3 \left(2 - \zeta_1 - \zeta_2\right)\left(\zeta_1 + \zeta_2\right)} \,, \qquad \delta y = \pi(1 - \zeta_2) \,, \qquad (3.6)$$

where $\alpha$ is

$$\alpha^2 = 2\pi^4 k \left(1 - \zeta_1^2\right)\left(1 - \zeta_2^2\right) \,. \qquad (3.7)$$

Finally, the free energy in the mass-deformed case is found

$$F = \frac{\pi}{3} \sqrt{2k \left(1 - \zeta_1^2\right)\left(1 - \zeta_2^2\right)} \, N^{3/2} \,, \qquad (3.8)$$

exhibiting the expected $N^{3/2}$ scaling.

## 3.2 The Wilson loop

We are now ready to exploit the previous results to compute BPS Wilson loops in the mass-deformed background, at leading order in the M-theory limit. According to the rule of (3.1), we have that

$$\langle W^{1/6} \rangle = \frac{2\pi i N}{k} \int_C dx \, \rho(x) e^{\lambda(x)} \,, \qquad \langle W^{1/2} \rangle = \frac{2\pi i N}{k} \int_C dx \, \rho(x) \left(e^{\lambda(x)} + e^{\mu(x)}\right) \,, \quad (3.9)$$

where the integrals extend over the support of the eigenvalue distribution, and $\lambda(x), \mu(x)$ denote the continuous limit of the eigenvalue distribution (3.2). In the superconformal case, the prescription successfully reproduced the leading behavior of the BPS Wilson loop at strong coupling [37].

We perform here an analogous computation with the masses $\zeta_1$, $\zeta_2$ turned on. Unlike the superconformal case, we are not aware of an explicit form for the imaginary part of the functions $y^{(1)}(x)$ and $y^{(2)}(x)$, as the extremization of the free energy fixes only their difference $\delta y$. However, the imaginary part is subleading in $N$ with respect to the real part

and, consequently, it cannot affect the leading exponential behavior of the Wilson loop: we can still obtain a reliable estimate of its mass dependence from the computation of (3.9).

The explicit result for the 1/6–BPS Wilson loop reads

$$\langle W^{1/6} \rangle = \frac{i}{\pi(\zeta_1 + \zeta_2)} \left[ \frac{(1-\zeta_2)}{(2-\zeta_1-\zeta_2)} e^{\frac{\pi\sqrt{2N(1-\zeta_1^2)(1-\zeta_2^2)}}{(1-\zeta_2)\sqrt{k}}} - \frac{(1+\zeta_1)}{(\zeta_1+\zeta_2+2)} e^{\frac{\pi\sqrt{2N(1-\zeta_1^2)(1-\zeta_2^2)}}{(\zeta_1+1)\sqrt{k}}} \right],$$
(3.10)

and, as expected, we observe in the exponential factor a peculiar dependence on the mass parameters[5].

We remark the emergence of a novel double exponential behavior in (3.10), which may signal a rich underlying structure depending on the relative values of $\zeta_1$ and $\zeta_2$. We recover, instead, a single exponential in two limits. The first one is when $\zeta_2 = -\zeta_1 = \zeta$, where we can interpret the double deformation as a single imaginary FI deformation. In this case the explicit result is

$$\langle W^{1/6} \rangle = i(1+\zeta)\sqrt{\frac{N}{2k}} \, e^{\frac{\sqrt{2}\pi(\zeta+1)\sqrt{N}}{\sqrt{k}}} \, .$$
(3.11)

Nicely, up to a suitable redefinition of the deformation parameter the dependence on $\zeta$ agrees with that found in [49][6].

The other special point is the superconformal limit of vanishing masses. Apparently (3.10) seems singular in such a limit, but a more careful analysis shows that is finite [7] leading to

$$\langle W^{1/6} \rangle \sim \sqrt{\frac{N}{k}} e^{\frac{\sqrt{2}\pi\sqrt{N}}{\sqrt{k}}} \, ,$$
(3.12)

in perfect agreement with [37].

In the next section we propose an alternative, and perhaps more solid, derivation of this result, which incorporates all the perturbative correction in $1/N$. We anticipate that the two methods give the same leading exponential for the Wilson loop, confirming all the features we observed above.

## 4    The large $N$ limit from the Fermi gas method

The strong coupling limit of ABJM-like models is particularly rich, with different behaviours corresponding to different scalings of $k$ and $N$. Various approaches have been developed [33, 35, 37] along the years and sometimes they can be extended to the massive case [47–49, 69–72]. Among them, the most fruitful is perhaps the Fermi gas method [33]. The reason is that corrections in $1/N$, perturbative and non-perturbative, are naturally encoded in this formalism. We shall focus on perturbative corrections, and defer the study of non-perturbative contributions to future investigations.

---

[5]We do not compute the vev of the 1/2–BPS Wilson loop as its differences from the bosonic one lies in the imaginary parts of the eigenvalues, which are unknown.

[6]The precise identification is $\zeta_{\text{here}} = \frac{4i\zeta_{\text{there}}}{k}$.

[7]See also Appendix B for the analogous computation which keeps into account also $1/N$ perturbative corrections

The original observation [33] is that the ABJM matrix model can be written as a statistical system of $N$ one-dimensional non-interacting fermions. Several generalizations were later proposed, including the possibility of turning on the massive background described in the previous sections [47]. One appealing feature of the Fermi gas approach is the correspondence between the $1/N$ expansion of ABJM and the quantum corrections of the statistical model, due to a proportionality relation between the Chern-Simons coupling and the Planck constant of the gas. Since quantum corrections are accessible via somewhat standard WKB techniques, the method gives a powerful tool to explore the strong coupling dynamics beyond the results of Section 3.

Even more interestingly, one can incorporate in the Fermi gas formalism the presence of quantum operators, both local [53, 73, 74] and extended, like BPS Wilson loops [42]. In the latter case, without mass deformations, it is possible to compute all the $1/N$ corrections and some non-perturbative contributions [44]. In the following, we will generalize the results of [42] to the presence of the mass terms $\zeta_1$ and $\zeta_2$.

Before entering the details, which might result technical for a first reading, we would like to present the main results of our computations. We obtained the exact expression for the expectation value of the $n$-winding 1/6–BPS Wilson loop in the mass-deformed background of (2.2)

$$
\langle W_n^{1/6} \rangle = \tilde{B}_1(k, \zeta_1, \zeta_2) \frac{\text{Ai}\left[ C^{-1/3} \left( N - B - \frac{2n}{k(1-\zeta_2)} \right) \right]}{\text{Ai}\left[ C^{-1/3} \left( N - B \right) \right]} +
$$
$$
+ \tilde{B}_2(k, \zeta_1, \zeta_2) \frac{\text{Ai}\left[ C^{-1/3} \left( N - B - \frac{2n}{k(1+\zeta_1)} \right) \right]}{\text{Ai}\left[ C^{-1/3} \left( N - B \right) \right]} ,
\tag{4.1}
$$

with

$$
C = \frac{2}{\pi^2 k (1 - \zeta_1^2)(1 - \zeta_2^2)} \quad , \quad B = \frac{k}{24} + \frac{2 + \zeta_1^2 + \zeta_2^2}{6k(1 - \zeta_1^2)(1 - \zeta_2^2)} ,
$$
$$
\tilde{B}_1(k, \zeta_1, \zeta_2) = \frac{i^{n(1-\zeta_2)}}{2\pi n! (2 - \zeta)} \Gamma\left( 1 + n - \frac{n\zeta}{2} \right) \Gamma\left( n\frac{\zeta}{2} \right) \csc\left( \frac{2n\pi}{k(1 - \zeta_2)} \right) ,
\tag{4.2}
$$
$$
\tilde{B}_2(k, \zeta_1, \zeta_2) = \frac{i^{n(1+\zeta_1)}}{2\pi n! (2 + \zeta)} \Gamma\left( 1 + n + \frac{n\zeta}{2} \right) \Gamma\left( -n\frac{\zeta}{2} \right) \csc\left( \frac{2n\pi}{k(1 + \zeta_1)} \right) ,
$$

where $\zeta \equiv \zeta_1 + \zeta_2$. From that, we can easily compute the vev of the $\frac{1}{2}$–BPS Wilson loop

$$
\langle W_n^{1/2} \rangle = \tilde{B}_1(k, \zeta_1, \zeta_2) \left( 1 - (-1)^{-n\zeta_2} \right) \frac{\text{Ai}\left[ C^{-1/3} \left( N - B - \frac{2n}{k(1-\zeta_2)} \right) \right]}{\text{Ai}\left[ C^{-1/3} \left( N - B \right) \right]} +
$$
$$
+ \tilde{B}_2(k, \zeta_1, \zeta_2) \left( 1 - (-1)^{n\zeta_1} \right) \frac{\text{Ai}\left[ C^{-1/3} \left( N - B - \frac{2n}{k(1+\zeta_1)} \right) \right]}{\text{Ai}\left[ C^{-1/3} \left( N - B \right) \right]} .
\tag{4.3}
$$

The expansion at large $N$ of these results agrees, at leading order, with that we have derived from the eigenvalue distribution, providing a nice cross check.

In the rest of the section, we present an introduction to the necessary technology of the Fermi gas, that can be skipped by people already familiar with this methodology. We

focus on two relevant examples: the ABJM partition function in the presence of mass deformations $\zeta_1$ and $\zeta_2$ and the computation of Wilson loops in the superconformal ABJM model.

## 4.1 A crash course on the Fermi gas formalism

To begin with, we rearrange the matrix model of mass-deformed ABJM (2.3) in a more convenient form. After some manipulations detailed, e.g., in [47], the partition function can be written as

$$Z(N) = \frac{1}{N!} \prod_{i=1}^{N} \int \frac{d\lambda_i}{2\pi} \ \det_{ij} \rho_0(\lambda_i, \lambda_j) = \frac{1}{N!} \prod_{i=1}^{N} \int \frac{dx_i}{2\pi} \ \det_{ij} \langle x_i | \hat{\rho} | x_j \rangle , \qquad (4.4)$$

where we set $x_i = k\lambda_i$, and promoted it to be the eigenvalue of an auxiliary position operator $\hat{x}_i$, whose physical interpretation will be clarified soon. We also have implicitly introduced the canonical momentum operator $\hat{p}_i$ and the corresponding eigenstates $|x_i\rangle, |p_i\rangle$ respectively. They satisfy the standard quantization condition

$$[\hat{x}_i, \hat{p}_j] = i\hbar\delta_{ij} = 2\pi i k \delta_{ij} , \qquad (4.5)$$

where $\hbar$ is not the physical Planck constant but rather a convenient interpretation of the Chern-Simons level. Then, we recognize (4.4) as the partition function of a quantum Fermi gas of $N$ non-interacting particles, with position $x_i$. Accordingly, $\hat{\rho}$ corresponds to the one-particle density matrix for such a gas and, after performing a unitary transformation to bring it into an Hermitian form, its expression reads as

$$\hat{\rho} = \frac{e^{-\frac{\zeta_1}{4}\hat{Q}}}{\left(2\cosh\frac{\hat{Q}}{2}\right)^{\frac{1}{2}}} \frac{e^{\frac{\zeta_2}{2}\hat{P}}}{2\cosh\frac{\hat{P}}{2}} \frac{e^{-\frac{\zeta_1}{4}\hat{Q}}}{\left(2\cosh\frac{\hat{Q}}{2}\right)^{\frac{1}{2}}} , \qquad (4.6)$$

where the operators $\hat{Q}$ and $\hat{P}$ are related to $\hat{x}$ and $\hat{p}$ through a suitable canonical transformation [47]. Finally, we define a quantum Hamiltonian $\hat{H}$ from $\hat{\rho} = e^{-\hat{H}}$: neglecting all the $\hbar$ corrections, $\hat{H}$ can be written as a sum of kinetic and potential term, $\hat{T}(P)$ and $\hat{U}(Q)$ respectively

$$\hat{T}(P) = -\frac{\zeta_2}{2}\hat{P} + \log\left(2\cosh\frac{\hat{P}}{2}\right) , \qquad \hat{U}(Q) = \frac{\zeta_1}{2}\hat{Q} + \log\left(2\cosh\frac{\hat{Q}}{2}\right) . \qquad (4.7)$$

The quantum Hamiltonian, of course, has not the simple form $T(P) + U(Q)$. Quite interestingly, we see that the effect of the $R-$charge deformations is a shift of the kinetic and the potential terms, recovering easily the superconformal ABJM case [33] at this level.

Remarkably, the vevs of 1/6–BPS Wilson loops, and hence also of the fermionic ones through the relation (2.13), admit a natural reinterpretation within this framework as 1-body operators [42]. A 1-body operator $\hat{\mathcal{O}}_1$ is an operator invariant under permutations and acting on states $|x_1, \dots, x_N\rangle$ of the Hilbert space of $N$ distinguishable particles as

$$\hat{\mathcal{O}}_1 |x_1, \dots, x_N\rangle = \sum_{i=1}^{N} \mathcal{O}_1(x_i) |x_1, \dots, x_N\rangle . \qquad (4.8)$$

In the case of fermionic particles, the states $|x_1, \ldots, x_N\rangle$ must be antisymmetrized according to the Pauli exclusion principle. The canonical thermal average of a 1-body operator can be expressed in terms of the single particle density matrix $\rho_1$ as

$$\langle \mathcal{O} \rangle_N = \frac{1}{n!} \int dx_1 \mathcal{O}_1(x_1) \rho_1(x) \qquad \rho_1(x_1) = \frac{1}{N!} \int dx_2 \ldots dx_N \rho(x_1, \ldots, x_N) \,, \qquad (4.9)$$

where $\langle \rangle_N$ denotes unnormalized vevs. Following the procedure described for the partition function, one finds that the insertion corresponding to the $n$-winding Wilson loop, in terms of the $Q$, $P$ variables, is [42]

$$\langle W^{1/6} \rangle_N = e^{\frac{n}{k}(Q+P)} \,. \qquad (4.10)$$

This representation holds also in the mass-deformed case, as we will show explicitly in the next section.

Now, the striking advantage of the Fermi gas formulation is the possibility to explore the strongly coupled regime of ABJM-like theories, i.e. $k \ll 1$, by performing a standard WKB expansion of the statistical model defined by (4.6). An efficient way to implement a systematic WKB expansion is to introduce the Wigner-Kirkwood formalism [33]. In this approach one associates to any operators $\hat{A}$ a quantum function on the phase space $A_W$ through the so-called Wigner transform

$$A_W(Q, P) = \int dQ' \, \langle Q - \frac{Q'}{2} | \hat{A} | Q + \frac{Q'}{2} \rangle \, e^{iPQ'/\hbar} \,. \qquad (4.11)$$

Then, statistical expectation values are written as

$$\langle A \rangle = \text{Tr} \, (\hat{\rho}\hat{A}) = \int \frac{dQ dP}{2\pi\hbar} \, A_W(Q, P) \star \rho_W(Q, P) \,, \qquad (4.12)$$

where $\rho_W$ is the Wigner transform of the density matrix. We also recall that the Wigner transform of the product of two operators is the Moyal product, denoted by $\star$, of the transformed quantities, namely

$$(\hat{A}\hat{B})_W = A_W \star B_W \equiv A_W \exp\left[ \frac{i\hbar}{2} (\overset{\leftarrow}{\partial_Q} \overset{\rightarrow}{\partial_P} - \overset{\leftarrow}{\partial_P} \overset{\rightarrow}{\partial_Q}) \right] B_W \,. \qquad (4.13)$$

It is not difficult to obtain the Wigner transform of the quantum Hamiltonian exploiting the Baker-Campbell-Hausdorff formula

$$\rho_W = e_\star^{-H_W} = e^{-\frac{1}{2}U(\hat{Q})} \star e^{-T(\hat{P})} \star e^{-\frac{1}{2}U(\hat{Q})} \,. \qquad (4.14)$$

We stress again that the full quantum Hamiltonian is not of the type $T + U$, but contains an infinite tower of corrections coming from nested commutators

$$H_W(Q, P) = T + U + \frac{1}{12}[T, [T, U]_\star]_\star + \frac{1}{24}[U, [T, U]_\star]_\star + \cdots =$$
$$= T(P) + U(Q) - \frac{\hbar^2}{12}(T'(P))^2 U''(Q) + \frac{\hbar^2}{24}(U'(Q))^2 T''(P) + \mathcal{O}(\hbar^4) \,. \qquad (4.15)$$

Up to now, we always have worked at fixed $N$: as observed in the seminal paper [33], it is convenient, for the large $N$ limit, to evaluate thermodynamic quantities in the grand-canonical ensemble. The grand-canonical partition function $\Xi$ is therefore constructed as

$$\Xi(\mu) = 1 + \sum_{N=1}^{\infty} Z(N) e^{N\mu} = e^{J(\mu)}, \tag{4.16}$$

being $\mu$ the chemical potential, $z \equiv e^{\mu}$ the fugacity and defining implicitly the grand-canonical potential $J(\mu)$. Given $\Xi$, the canonical partition function is recovered from

$$Z(N) = \oint \frac{dz}{2\pi i} \frac{\Xi(\log z)}{z^{N+1}} = \frac{1}{2\pi i} \int d\mu\, \Xi(\mu)\, e^{-\mu N}. \tag{4.17}$$

An analogous prescription applies to observables. Specifically, for the Wilson loop

$$\langle W^{1/6} \rangle_{\mathrm{GC}} = \sum_{N=1}^{\infty} \langle W^{1/6} \rangle_N z^N, \qquad \langle W^{1/6} \rangle_N = \frac{1}{2\pi i} \int d\mu\, e^{-\mu N} \langle W^{1/6} \rangle_{\mathrm{GC}}. \tag{4.18}$$

We can finally present the results for the partition function in the presence of mass deformations. The relevant point is to compute $J(\mu)$: an useful approach is to consider the distribution operator $\hat{n}(E)$, which counts eigenstates with energy less than $E$ and is defined as

$$\hat{n}(E) = \mathrm{Tr}\ \theta(E - \hat{H}). \tag{4.19}$$

Its explicit relation with the grand potential is

$$J(\mu) = \int dE \frac{dn}{dE} \log\big(1 + z e^{-E}\big), \tag{4.20}$$

and building on this formula, one can argue [48] that

$$J(\mu) = \frac{C}{3}\mu^3 + B\mu + A + \mathcal{O}(e^{-\mu}), \tag{4.21}$$

with $A$, $B$ and $C$ being functions of $k, \zeta_1, \zeta_2$, generalizing the result of the undeformed case [33]. The $\mathcal{O}(e^{-\mu})$ refers to all the non-perturbative terms. Up to these exponentially small corrections, plugging (4.21) into (4.17) and performing the integral, one express $Z(N)$ as an Airy function

$$Z(N) = e^A\, C^{-1/3}\, \mathrm{Ai}[C^{-1/3}(N - B)]. \tag{4.22}$$

This expression encodes a resummation of all the perturbative $1/N$ corrections, but is valid for any value of $k$, up to exponentially small terms, that correspond to instanton effects in M-theory.

The coefficients $B$ and $C$ are obtained from the perturbative part of $n(E)$, accessible via the WKB expansion. One can efficiently derive them by expanding the distribution operator, seen as a function of the Hamiltonian $\hat{H}$, around $H_W(Q, P)$ and then taking its Wigner transform. Explicitly, one gets

$$n(E) = \mathrm{Tr}\ \hat{n}(E)_W = \int_{H_W(Q,P) \leq E} \frac{dQ\,dP}{2\pi\hbar} + \sum_{r=1}^{\infty} \frac{1}{r!} \int \frac{dQ\,dP}{2\pi\hbar} \mathcal{G}_r \delta^{(r-1)}(E - H_W), \tag{4.23}$$

where

$$\mathcal{G}_r = \left[ (\hat{H} - H_W(q,p))^r \right]_W . \tag{4.24}$$

The above expression has a simple physical interpretation: the first term accounts for the quantum corrections to the Fermi surface, namely the region in the phase space such that $H_W(P,Q) \leq E$, due to the $\hbar$ corrections to the Hamiltonian, and the second is the standard semiclassical expansion of the density of eigenvalues.

The coefficient $C$ can be easily derived in the thermodynamic limit: it is only sensible to the leading term in $n(E)$ that is proportional to the semiclassical volume of the Fermi surface, which goes as $E^2$. Perturbative WKB corrections in $Z$ are of order $(\hbar \frac{\mathrm{d}}{\mathrm{d}E})^2$, and cannot affect the leading behavior, so $C$ is left untouched. Following the same logic, they affect $B$, but only $\hbar$ corrections to the Hamiltonian up to the second order play a role. An explicit calculation leads to the expressions of (4.2) for $B$ and $C$ [47]. The computation of $A$ is instead sensible to the terms of order $e^{-E}$ into $n(E)$. However, building on numerical results and analogy with topological string theory, an exact expression was found [47, 75], that is

$$A = \frac{1}{4} \left( \mathcal{A}_{\mathrm{ABJM}}(k(1+\zeta_1)) + \mathcal{A}_{\mathrm{ABJM}}(k(1-\zeta_1)) + \mathcal{A}_{\mathrm{ABJM}}(k(1+\zeta_2)) + \mathcal{A}_{\mathrm{ABJM}}(k(1-\zeta_2)) \right) , \tag{4.25}$$

with $\mathcal{A}_{\mathrm{ABJM}}$ the constant map function

$$\mathcal{A}_{\mathrm{ABJM}}(k) = \frac{2\zeta(3)}{\pi^2 k} \left( 1 - \frac{k^3}{16} \right) + \frac{k^2}{\pi^2} \int_0^\infty dx \frac{x}{e^{kx} - 1} \log\left( 1 - e^{-2x} \right) = \tag{4.26}$$

$$= 2\zeta'(-1) - \frac{1}{6} \log \frac{k}{4\pi} - \frac{\zeta(3)}{8\pi^2} k^2 + \frac{1}{3} \int_0^\infty \frac{dx}{e^{kx} - 1} \left( \frac{3}{x \sinh^2 x} - \frac{3}{x^2} + \frac{1}{x} \right) .$$

Let us now discuss the case of Wilson loops in the superconformal limit, i.e., in the absence of mass deformations [42]. The relevant quantities can be read from the deformed case by setting to zero $\zeta_1$ and $\zeta_2$. We use the following expression for the single particle grand-canonical partition function

$$\rho_{\mathrm{GC}} = \frac{\Xi}{e^{\hat{H}-\mu} + 1} = \Xi \pi \partial_\mu \left[ \csc(\pi \partial_\mu) \theta(\mu - \hat{H}) \right] , \tag{4.27}$$

which also implies

$$\langle \mathcal{O} \rangle_{\mathrm{GC}} = \Xi \, \mathrm{Tr} \left( \frac{\mathcal{O}}{e^{\hat{H}-\mu} + 1} \right) \equiv \Xi \pi \partial_\mu \csc(\pi \partial_\mu) n_{\mathcal{O}}(\mu) , \tag{4.28}$$

where $n_{\mathcal{O}}(\mu)$ is implicitly defined by (4.28). Its explicit form in the language of the Wigner transform, which is the most convenient for concrete calculations, is

$$n_{\mathcal{O}_n}(\mu) = \int \frac{dQ dP}{2\pi\hbar} \, \theta(\mu - H_W) \, e^{\frac{n(Q+P)}{k}} + \sum_{r \geq 1} \frac{(-1)^r}{r!} \frac{d^{r-1}}{d\mu^{r-1}} \int \frac{dQ dP}{2\pi\hbar} \, \delta(\mu - H_W) \, \mathcal{G}_r \, e^{\frac{n(Q+P)}{k}} , \tag{4.29}$$

where again we expanded $n_{\mathcal{O}}(\mu)$ around $H_W$. We stress that, unlike the case of the partition function, there is a technical complication because one has to consider all the $\hbar$ corrections. That is, we have to consider an infinite series of terms of the form

$$U^{(n)}(Q)(T'(P))^n \quad , \quad T^{(n)}(P)(U'(Q))^n . \tag{4.30}$$

Nevertheless, in the superconformal case the computation for the Wilson loop can be carried on [42] and leads to

$$n_{\mathcal{O}_n}(\mu) = \frac{k}{2\pi n\hbar} \, e^{\frac{2n\mu}{k}} \, i^n \left( 2\mu - \frac{i\pi k}{2} - kH_n \right) , \qquad (4.31)$$

where $H_n$ is the $n$-th harmonic number. It is not hard to see that this expression, after some manipulations leads again to a combination of Airy functions and its derivative. However, we do not need those explicit results here. The interested reader can find them in the Appendix B, where we discuss the zero mass limit of the results we are about to derive.

## 5 Wilson loops in the mass-deformed theory

We are ready now to obtain the explicit expressions (4.1) and (4.3) for the BPS Wilson loops in the mass-deformed theory. As explained before, we know that calculating the vev of $1/6$-BPS Wilson loop with winding number $n$ in the Fermi gas approach corresponds to inserting

$$\mathcal{O}_n = \sum_{i=1}^{n} e^{n\lambda_i} , \qquad (5.1)$$

in the matrix integral. As shown before, we pose $\lambda_i = \frac{x_i}{k}$ and we have that $Q + P = x$: the vev of $1/6$-BPS Wilson loop is obtained by inserting the operator

$$\mathcal{O}_n = \exp\left( \frac{n\,x}{k} \right) = \exp\left( \frac{n(Q+P)}{k} \right) . \qquad (5.2)$$

It is interesting, at this point, to remark on the difference between the present computation and the one for the latitude Wilson loops in ABJM [24]: in that case, the deformation parameter modifies the operator itself representing the Wilson loop. On the contrary, here, the operator is unchanged, while the bulk theory is deformed.

In practice, we extend the derivation of [42] to the presence of the real masses $\zeta_1, \zeta_2$. Even if the underlying logic is similar, we report the main steps of the computations, emphasizing the differences due to the presence of the masses. Interested readers can find additional technical details in the Appendix A.

### 5.1 Warm up: recovering the leading exponential term

As a first step, to fix the ideas and to compare against the findings of Section 3, we shall examine the strict large $N$ limit. The regime $N \to \infty$ corresponds to the standard thermodynamic limit of the statistical system, where the Hamiltonian takes the rather simple form

$$H_W(Q, P) = \frac{|P| - \zeta_2 P}{2} + \frac{|Q| + \zeta_1 Q}{2} , \qquad (5.3)$$

that makes straightforward to evaluate $n_{\mathcal{O}_n}(\mu)$ directly from the first term of (4.29). We plug the above expression into (4.28), to find the grand canonical vev of the $1/6$–BPS

Wilson loop. The explicit result reads as

$$\frac{1}{\Xi}\langle O_n\rangle_{\text{GC}} = \frac{e^{\frac{2n\mu}{k(1-\zeta_2)}}}{n\pi\zeta(2-\zeta)}\csc\left(\frac{2n\pi}{k(1-\zeta_2)}\right) - \frac{e^{\frac{2n\mu}{k(1+\zeta_1)}}}{n\pi\zeta(2+\zeta)}\csc\left(\frac{2n\pi}{k(1+\zeta_1)}\right), \qquad (5.4)$$

where we recall that $\zeta \equiv \zeta_1 + \zeta_2$. Next, we can compute the canonical vev in the thermodynamic limit by performing a standard saddle-point approximation in (4.18), now suitably normalized. It is enough to evaluate the grand canonical vev (5.4) for the value $\mu_\star$, determined by requiring that the grand canonical average $N(\mu)$, seen as a function of $\mu$, is equal to the canonical constant value $N$

$$\mu_* = \pi\sqrt{\frac{kN}{2}}\sqrt{(1-\zeta_1^2)(1-\zeta_2^2)}. \qquad (5.5)$$

We substitute $\mu_\star$ into (5.4) and find

$$\langle W_n^{1/6}\rangle \approx e^{\frac{n\pi}{1-\zeta_2}\sqrt{\frac{2N}{k}}\sqrt{(1-\zeta_1^2)(1-\zeta_2^2)}} \frac{\csc\left(\frac{2n\pi}{k(1-\zeta_2)}\right)}{n\pi\zeta(2-\zeta)} - e^{\frac{n\pi}{1+\zeta_1}\sqrt{\frac{2N}{k}}\sqrt{(1-\zeta_1^2)(1-\zeta_2^2)}} \frac{\csc\left(\frac{2n\pi}{k(1+\zeta_1)}\right)}{n\pi\zeta(2+\zeta)}. \tag{5.6}$$

We finally compare this expression with (3.10), obtained independently from continuous eigenvalue distribution. Even if the computations of the prefactors are not fully reliable in both cases [8], we observe an exact correspondence in the leading exponential behaviour. We think that the matching of the two exponential factors, obtained from two unrelated techniques, is a non-trivial cross-check of our results.

## 5.2 The full perturbative computation: the number of states

We face the more challenging problem of computing and resumming all the perturbative quantum $1/N$ corrections, neglecting the exponentially small factors. The central goal of this section is the explicit computation of (4.29). Since the task is somewhat involved, we divide the computation into different steps.

### 5.2.1 The number of states: part I

We begin with some preliminary observations about the structure of the integrals to be evaluated. First of all, we split (4.29) in two contributions, namely

$$n_{\mathcal{O}_n}^{(1)}(\mu) = \int \frac{dQ\,dP}{2\pi\hbar}\,\theta(\mu - H_W)\,e^{\frac{n(Q+P)}{k}},$$
$$n_{\mathcal{O}_n}^{(2)}(\mu) = \sum_{r\geq 1}\frac{(-1)^r}{r!}\frac{d^{r-1}}{d\mu^{r-1}}\int \frac{dQ\,dP}{2\pi\hbar}\,\delta(\mu - H_W)\,\mathcal{G}_r\,e^{\frac{n(Q+P)}{k}}. \tag{5.7}$$

They represent, respectively, the integration of the operator that corresponds to the Wilson loop (5.2) over the Fermi surface and over the quantum corrected boundary of the Fermi

---

[8]Expanding the full result (4.1), we find a mismatch even with the massless case of [42]. Presumably, the thermodynamic argument is too naive to reproduce the coefficients of the exponentials. For (3.10), see the discussion in the corresponding section.

surface, defined by $H_W(q,p) = \mu$. Then, following [33], it is useful to split the Fermi surface into regions where the semiclassical approximation either for $T(p)$ or $U(q)$ makes sense. For instance, if we select a region where $P$ is of order $\mu$, we can approximate $T(P)$ in (4.7) with its leading term for large $P$, i.e.

$$T(P) \simeq \frac{|P| - \zeta_2 P}{2}\,. \tag{5.8}$$

In turn, the quantum Hamiltonian restricted to that region takes the form [42]

$$H_W(Q,P) = T(P) + U(Q) - \frac{i\hbar}{2}T'(P)U'(Q) + \sum_{m \geq 1} \frac{B_{2m}}{(2m)!}(i\hbar)^{2m}U^{(2m)}(Q)(T'(P))^{2m}\,, \tag{5.9}$$

where $B_{2m}$ are the Bernoulli numbers. The important observation is that higher order corrections in (5.9), apart from the first one in $\hbar$, contain terms that are exponentially suppressed in the large $\mu$ limit, as for large $Q$ the derivatives of $U(Q)$ behave as

$$U^{(2m)}(Q) = O(e^{-\mu})\,, \qquad m \geq 1\,. \tag{5.10}$$

We can neglect all these contributions in many parts of the computation. Using the same arguments, in the region where $Q \sim \mu$, it is possible to argue that the Hamiltonian is corrected only by terms of the form $(U'(Q))^{2m}T^{(2m)}(P)$ and that those corrections are exponentially small for $m \geq 1$.

Of course, the delicate point is to correctly identify the relevant regions. In the massless case, they were expressed in terms of a point of the massless Fermi surface with $P \sim \mu + O(\hbar)$ and $Q \sim \mu + O(\hbar) + O(e^{-\mu})$. We modify that ansatz, and select a special point on the Fermi surface imposing the additional constraint that it must respect the symmetry for the exchange $\zeta_1 \leftrightarrow -\zeta_2$. The most natural point $(Q_*, P_*)$ with these properties in our case is[9]

$$P_* = \frac{\mu}{1 - \zeta_2} + \frac{i\hbar}{8}(1 + \zeta_1)\,, \quad Q_* = \frac{\mu}{1 + \zeta_1} + \frac{i\hbar}{8}(1 - \zeta_2) + \mathcal{O}(e^{-\mu})\,. \tag{5.11}$$

In the massless limit, we smoothly recover the choice of [42]. See Fig. 1 for a graphical representation, to which we refer also in the following considerations.

In the first quadrant, the point $(Q_*, P_*)$ divides the boundary of the Fermi surface into two segments, selecting regions $a$ and $b$ whose area is

$$Vol_a = \int_0^{Q_*} P(\mu, Q)\, dQ \quad , \quad Vol_b = \int_0^{P_*} Q(\mu, P)\, dP - P_* Q_*\,, \tag{5.12}$$

and where $P(\mu, Q)$ and $Q(\mu, P)$ are local solutions of the approximated quantum Hamiltonian, defined by

$$H_W = T(P) + U(Q) - \frac{i\hbar}{2}U'(Q)T'(P) + \mathcal{O}(\hbar^2)\,. \tag{5.13}$$

---

[9]As in [42], we perform a Wick rotation such that $i\hbar$ is real.

The boundary of the region $a$ is determined by $P \geq P_*$, that is $P(\mu, Q) \geq \frac{\mu}{1-\zeta_2}$, while on the boundary of region $b$ we have that $Q(\mu, P) \geq \frac{\mu}{1+\zeta_1}$. Extending the very same argument on the other quadrants, we can define two further regions: region I as

$$P > P_*, \qquad -\tilde{Q}_* \leq Q \leq Q_*. \qquad (5.14)$$

and region II as

$$Q > Q_*, \qquad -\tilde{P}_* \leq P \leq P_*. \qquad (5.15)$$

where the exact expressions of $\tilde{Q}_*$ and $\tilde{P}_*$ are not relevant to our computation. In these regions, we can safely assume

$$\begin{aligned} \text{region} \ \ \text{I}: \quad & e^{-P} < e^{-\frac{\mu}{1-\zeta_2}}, \\ \text{region} \ \ \text{II}: \quad & e^{-Q} < e^{-\frac{\mu}{1+\zeta_1}}. \end{aligned} \qquad (5.16)$$

Namely, exponentially small terms in $P$ and $Q$ are bounded by exponentially small terms in $\mu$ and we can neglect terms of the form $T^{(m)}(P)$ and $U^{(m)}(Q)$ (for $m \geq 2$) in regions I and II, respectively. As proven in [33] and explained above, contributions of the form $U^{(m)}(Q)$ and $T^{(m)}(P)$ give exponentially small corrections in regions I and II, respectively. In the end, the relevant part of the integral can be evaluated by taking into account the contributions from two new regions, denoted by $A$ and $B$, and then subtracting their complement in the bulk region that was overcounted. The two new regions are defined as

$$\begin{aligned} \text{region} \ \ A: \quad & P > 0 \quad, \quad -\tilde{Q}_* \leq Q \leq Q_*, \\ \text{region} \ \ B: \quad & Q > 0 \quad, \quad -\tilde{P}_* \leq P \leq P_*, \\ \text{bulk region}: \quad & -\tilde{P}_* \leq P \leq P_* \quad, \quad -\tilde{Q}_* \leq Q \leq Q_*. \end{aligned} \qquad (5.17)$$

Examining the Fermi surface in Fig. 1, it is evident that the overcounted region can be confined to the rectangle in the first quadrant, where $0 \leq P \leq P_*$ and $0 \leq Q \leq Q_*$. Indeed, any other additional contribution from the bulk of the Fermi surface introduces only negligible exponentially small terms.

To see these arguments in practice and make the above discussion concrete, we compute the contribution from region $A$. Here, the kinetic term can be written as

$$T(P) = \frac{P}{2}(1 - \zeta_2) + \mathcal{O}(e^{-\mu}), \qquad (5.18)$$

and the Fermi surface is defined by

$$\mu = H_W = \frac{P}{2}(1 - \zeta_2) + U(Q) - \frac{i\hbar}{4}(1 - \zeta_2)U'(Q) + \mathcal{O}(e^{-\mu}). \qquad (5.19)$$

Solving for $P$ we get

$$P(\mu, Q) = \frac{2\mu}{(1 - \zeta_2)} - \frac{2U(Q)}{(1 - \zeta_2)} + \frac{i\hbar}{2}U'(Q) = \frac{2\mu}{(1 - \zeta_2)} - \left( \frac{2H_W}{(1 - \zeta_2)} - P \right). \qquad (5.20)$$

In the first term $n_{\mathcal{O}_n}^{(1)}(\mu)$ of (5.7), the $\theta(\mu - H_W)$ implies that the integration is restricted to the region

$$H_W \leq \mu \quad \Leftrightarrow \quad P \leq P(\mu, Q). \qquad (5.21)$$

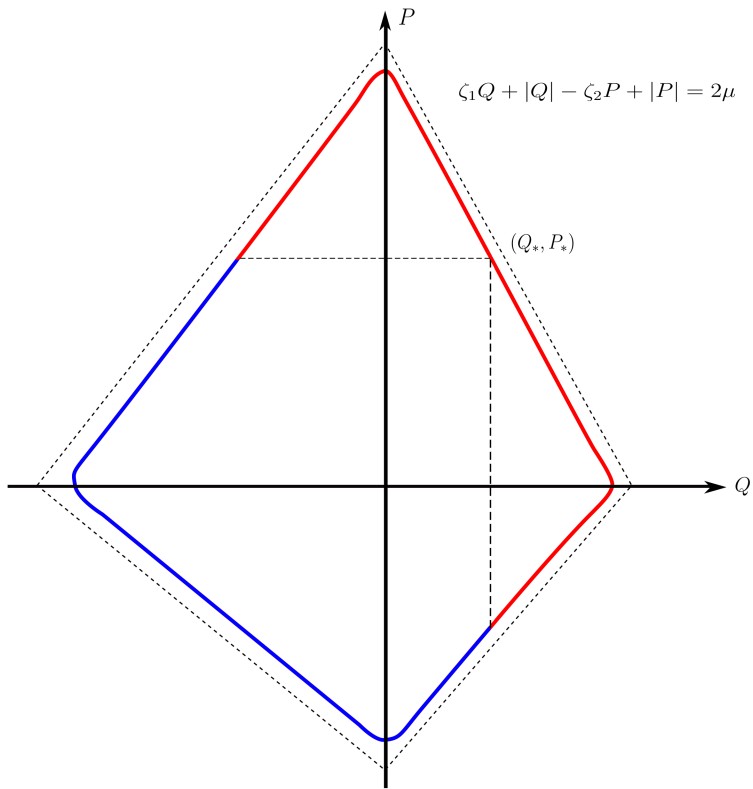

$$\zeta_1 Q + |Q| - \zeta_2 P + |P| = 2\mu$$

$(Q_*, P_*)$

**Figure 1**. The quantum Fermi surface

Therefore, keeping only the leading terms, we get from the region $A$

$$n_{\mathcal{O}_n}^{(1,A)}(\mu) = \int_{-\tilde{Q}_*}^{Q_*} \frac{dQ}{2\pi\hbar} e^{\frac{nQ}{k}} \int_0^{P(\mu,Q)} dP \, e^{\frac{nP}{k}} = \frac{k}{2\pi n\hbar} e^{\frac{2n\mu}{k(1-\zeta_2)}} \int_{-\tilde{Q}_*}^{Q_*} dQ \, e^{\frac{n(Q+P)}{k}} e^{-\frac{2n}{k(1-\zeta_2)}H_W} .$$
(5.22)

The evaluation of $n_{\mathcal{O}_n}^{(2,A)}(\mu)$ is more subtle as it is sensible to the infinite tower of higher order corrections of $H_W$. We can solve this problem by implementing the following identity

$$\delta(\mu - H_W(Q,P)) = \frac{1}{\left|\frac{\partial H_W(Q,P)}{\partial P}\right|} \delta(P - P(\mu,Q)) = \frac{2}{1-\zeta_2} \delta(P - P(\mu,Q)),$$
(5.23)

into (5.7). For instance, in region $A$, we can integrate over $P$ and the expressions for $n_{\mathcal{O}_n}^{(2)}(\mu)$ for region A reduces to

$$n_{\mathcal{O}_n}^{(2,A)}(\mu) = \frac{k}{2n\pi\hbar} e^{\frac{2n\mu}{k(1-\zeta_2)}} \sum_{r\geq 1} \frac{(-1)^r}{r!} \left(\frac{2n}{k(1-\zeta_2)}\right)^r \int_{-\tilde{Q}_*}^{Q_*} dQ \, \mathcal{G}_r e^{\frac{n(Q+P)}{k}} e^{-\frac{2n}{k(1-\zeta_2)}H_W} .$$
(5.24)

Still, we need to compute efficiently the generating functional of the Wigner-Kirkwood corrections, namely

$$e_\star^{-tH_W} = (e^{-t\hat{H}})_W = \left(\sum_{r=0}^{\infty} \frac{(-t)^r}{r!} \mathcal{G}_r\right) e^{-tH_W} .$$
(5.25)

Extending the regularization introduced in [42], we found the explicit expression in Appendix A for the specific value $t = \frac{2n}{k(1-\zeta_2)}$. The result is

$$e_\star^{-\frac{2n}{k(sgn(P)-\zeta_2)}H_W} = e^{-\frac{n}{k}|P| + \frac{n\pi i}{2}(1+\zeta_1) - \frac{n\zeta_1}{k(sgnP-\zeta_2)}Q - n \log\left(2\sinh\frac{Q}{k(sgnP-\zeta_2)}\right)}. \tag{5.26}$$

The sum in (5.24) can be safely done in terms of the generating functional (5.25) and (5.26), finally obtaining a compact formula for the number of eigenstates in the region A

$$n_{\mathcal{O}_n}^{(A)}(\mu) = \frac{k}{2n\pi\hbar} e^{\frac{2n\mu}{k(1-\zeta_2)}} e^{\frac{n\pi i}{2}(1+\zeta_1)} \int_{-\tilde{Q}_*}^{Q_*} dQ \, e^{\frac{nQ}{k}} \frac{e^{-\frac{n\zeta_1}{k(1-\zeta_2)}Q}}{\left(2\sinh\frac{Q}{k(1-\zeta_2)}\right)^n}. \tag{5.27}$$

### 5.2.2 The number of states: part II

The explicit evaluation of the integral (5.27) is non-trivial. As done in [42], we can send the lower integration limit to $-\infty$ up to exponentially small terms, that we are neglecting anyway. To simplify the computation, we make the substitution

$$u = e^{\frac{Q}{k(1-\zeta_2)}} \quad , \quad du = \frac{1}{k(1-\zeta_2)} e^{\frac{Q}{k(1-\zeta_2)}} dQ, \tag{5.28}$$

and we get

$$n_{\mathcal{O}_n}^{(A)}(\mu) = \frac{k^2}{2n\pi\hbar}(1-\zeta_2) \, e^{\frac{2n\mu}{k(1-\zeta_2)}} \, i^{n(1+\zeta_1)} \int_0^{u_*} du \, \frac{u^{n(1-\zeta_1-\zeta_2)-1}}{(u-u^{-1})^n}, \tag{5.29}$$

with the explicit expression of $u_*$ given by

$$u_* = \exp\left(\frac{Q_*}{k(1-\zeta_2)}\right) = \exp\left(\frac{\mu}{k(1+\zeta_1)(1-\zeta_2)} + \frac{i\pi}{4}\right). \tag{5.30}$$

It is useful to notice that the integral in (5.29) is of the form [24]

$$I_{a,b} = \int_0^{u_*} du \frac{u^{a-1}}{(u-u^{-1})^b} = \frac{(-1)^{-b}}{a+b} u_*^{a+b} \, _2F_1\left(b, \frac{a+b}{2}; \frac{a+b}{2}+1; u_*^2\right). \tag{5.31}$$

with $a = n(1-\zeta_1-\zeta_2)$ and $b = n$. In order to simplify our result, it is possible to remove exponentially subleading terms in $\mu$ exploiting identities for the hypergeometric functions. We use the identity

$$\begin{aligned}
_2F_1(\alpha,\beta,\gamma,z) &= \frac{(-z)^{-\alpha}\,\Gamma(\gamma)\Gamma(\beta-\alpha)}{\Gamma(\beta)\Gamma(\gamma-\alpha)} \, _2F_1(\alpha,\alpha-\gamma+1;\alpha-\beta+1;z^{-1}) + \\
&\quad + \frac{(-z)^{-\beta}\,\Gamma(\gamma)\Gamma(\alpha-\beta)}{\Gamma(\alpha)\Gamma(\gamma-\beta)} \, _2F_1(\beta,\beta-\gamma+1;-\alpha+\beta+1;z^{-1}),
\end{aligned} \tag{5.32}$$

and we expand each term in a power series: since we are neglecting exponentially small terms (i.e. terms $\propto z^{-1} = e^{-2\mu}$), we can consider only the first term in the expansion

$$_2F_1(\alpha,\beta,\gamma,z^{-1}) \approx 1 + \mathcal{O}(z^{-1}). \tag{5.33}$$

After some manipulations that involve identities of gamma functions, we obtain the final expression for the number of eigenstates in region $A$

$$n_{\mathcal{O}_n}^{(A)}(\mu) = \frac{k^2(1-\zeta_2)}{2n\pi\hbar} \; e^{\frac{2n\mu}{k(1-\zeta_2)}} \; i^{n(1+\zeta_1)} \left[ -\frac{1}{n\zeta} u_*^{-n\zeta} + \frac{(-1)^{\frac{n}{2}\zeta}}{n!(2-\zeta)} \; \Gamma\left(1+n-\frac{n\zeta}{2}\right) \Gamma\left(n\frac{\zeta}{2}\right) \right].$$

(5.34)

From symmetry considerations, the contribution from region $B$ is expected to be the same found for the region $A$ with $\zeta_1 \leftrightarrow -\zeta_2$. The explicit computation is a simple generalization of the one we have done before, and we recover

$$n_{\mathcal{O}_n}^{(B)}(\mu) = \frac{k^2(1+\zeta_1)}{2n\pi\hbar} \; e^{\frac{2n\mu}{k(1+\zeta_1)}} \; i^{n(1-\zeta_2)} \left[ \frac{1}{n\zeta} u_*^{n\zeta} + \frac{(-1)^{-\frac{n}{2}\zeta}}{n!(2+\zeta)} \; \Gamma\left(1+n+\frac{n\zeta}{2}\right) \Gamma\left(-n\frac{\zeta}{2}\right) \right].$$

(5.35)

As discussed before, to arrive at the final result, we should still subtract the contribution of the bulk region. Its computation is straightforward and gives

$$n_{\mathcal{O}_n}^{(bulk)}(\mu) = \int_{-\tilde{Q}_*}^{Q_*} \int_{-\tilde{P}_*}^{P_*} \frac{dQ\,dP}{2\pi\hbar} e^{\frac{n(Q+P)}{k}} \approx \frac{k^2}{2\pi\hbar n^2} e^{\frac{n}{k}(Q_*+P_*)} =$$

$$= \frac{k^2}{2\pi\hbar n^2} \exp\left[ \frac{n\mu}{k}\left(\frac{1}{1+\zeta_1} + \frac{1}{1-\zeta_2}\right) + \frac{in\hbar}{8k}(2+\zeta_1-\zeta_2) \right].$$

(5.36)

In conclusion, putting together all the different pieces [10], the leading expression of the total number of eigenstates can be rewritten in this form

$$n_{\mathcal{O}_n}(\mu) = A(k,\zeta_1,\zeta_2) \, e^{\frac{n\mu}{k}\left(\frac{1}{1+\zeta_1} + \frac{1}{1-\zeta_2}\right)} + B_1(k,\zeta_1,\zeta_2) \, e^{\frac{2n\mu}{k(1-\zeta_2)}} + B_2(k,\zeta_1,\zeta_2) \, e^{\frac{2n\mu}{k(1+\zeta_1)}} =$$

$$= B_1(k,\zeta_1,\zeta_2) \, e^{\frac{2n\mu}{k(1-\zeta_2)}} + B_2(k,\zeta_1,\zeta_2) \, e^{\frac{2n\mu}{k(1+\zeta_1)}},$$

(5.37)

since the first coefficient $A(k,\zeta_1,\zeta_2)$ is zero, while $B_1$ and $B_2$ are

$$B_1(k,\zeta_1,\zeta_2) = \frac{k^2}{2n\pi\hbar}(1-\zeta_2) \; \frac{i^{n(1-\zeta_2)}}{n!(2-\zeta)} \; \Gamma\left(1+n-\frac{n\zeta}{2}\right) \Gamma\left(n\frac{\zeta}{2}\right),$$

$$B_2(k,\zeta_1,\zeta_2) = \frac{k^2}{2n\pi\hbar}(1+\zeta_1) \; \frac{i^{n(1+\zeta_1)}}{n!(2+\zeta)} \; \Gamma\left(1+n+\frac{n\zeta}{2}\right) \Gamma\left(-n\frac{\zeta}{2}\right).$$

(5.38)

### 5.3 The full perturbative computation: $1/6-$BPS Wilson loop

Having obtained a formula for the number of states, we proceed with the calculation of the vev of the Wilson loop. We need to compute the r.h.s. of

$$\frac{1}{\Xi}\langle\mathcal{O}_n\rangle^{GC} = \pi\partial_\mu \csc(\pi\partial_\mu) n_{\mathcal{O}_n}(\mu).$$

(5.39)

Using the Taylor series of csc function, it is straightforward to attain the result

$$\frac{1}{\Xi}\langle\mathcal{O}_n\rangle^{GC} = B_1(k,\zeta_1,\zeta_2) \; \frac{2\pi n}{k(1-\zeta_2)} \; \csc\left(\frac{2n\pi}{k(1-\zeta_2)}\right) e^{\frac{2n\mu}{k(1-\zeta_2)}} +$$

$$+ B_2(k,\zeta_1,\zeta_2) \; \frac{2\pi n}{k(1+\zeta_1)} \; \csc\left(\frac{2n\pi}{k(1+\zeta_1)}\right) e^{\frac{2n\mu}{k(1+\zeta_1)}}.$$

(5.40)

---

[10] We choose the argument of $\zeta_1$, $\zeta_2$ to be such that $\log(-1) = -i\pi$. This reproduces the correct limit $\zeta_{1,2} \to 0$ and known results for the Bremsstrahlung we will present in Section 7.

As discussed before, we proceed with the calculation of the vev of the 1/6–BPS Wilson loop

$$\langle W_n^{1/6} \rangle = \frac{1}{2\pi i Z} \int d\mu \, e^{-\mu N} \langle \mathcal{O}_n \rangle^{GC} \,, \tag{5.41}$$

with $Z$ being the partition function found in (4.22). Substituting all the explicit expressions, the integral (5.41) is performed, giving a combination of Airy functions

$$\langle W_n^{1/6} \rangle = \tilde{B}_1(k,\zeta_1,\zeta_2) \, \frac{\text{Ai}\left[C^{-1/3}\left(N - B - \frac{2n}{k(1-\zeta_2)}\right)\right]}{\text{Ai}\left[C^{-1/3}\left(N - B\right)\right]} + \\ + \tilde{B}_2(k,\zeta_1,\zeta_2) \, \frac{\text{Ai}\left[C^{-1/3}\left(N - B - \frac{2n}{k(1+\zeta_1)}\right)\right]}{\text{Ai}\left[C^{-1/3}\left(N - B\right)\right]} \,, \tag{5.42}$$

where the relevant coefficients have been presented in (4.2). As expected, this vev has the same symmetry $\zeta_1 \leftrightarrow -\zeta_2$ of the matrix model. Notice that the non-perturbative term $A$, appearing in the partition function, simplifies against the same term in the normalization and the vev of the Wilson loop does not depend on it [44].

In the limit $\zeta_1, \zeta_2 \to 0$, the expression simplifies and we get the same result derived in [42]. The explicit computation is contained in Appendix A. We will comment on the structure of this result after having derived also the vev of the 1/2–BPS Wilson loop in the next section.

## 5.4 The full perturbative computation: $1/2-$BPS Wilson loop

To compute the vev of the fermionic Wilson loop, we exploit the formula (2.13). As explained in Section 2.2, the $\langle \hat{W}_n^{1/6} \rangle$ is computed from $\langle W_n^{1/6} \rangle$ simply by complex conjugation, even in the presence of mass deformations. The expression for the 1/6–BPS Wilson loop, $\langle \hat{W}_n^{1/6} \rangle$, is quickly obtained from (4.1) by taking the complex conjugate of the coefficients $\tilde{B}_1(k,\zeta_2,\zeta_1)$, $\tilde{B}_2(k,\zeta_2,\zeta_1)$, namely $\tilde{B}_1^*(k,\zeta_2,\zeta_1)$ and $\tilde{B}_2^*(k,\zeta_2,\zeta_1)$. Using that

$$(-1)^n \tilde{B}_1^*(k,\zeta_1,\zeta_2) = (-1)^{-n\zeta_2} \tilde{B}_1(k,\zeta_1,\zeta_2) \,, \\ (-1)^n \tilde{B}_2^*(k,\zeta_2,\zeta_1) = (-1)^{n\zeta_1} \tilde{B}_2(k,\zeta_1,\zeta_2) \,, \tag{5.43}$$

we arrive at

$$\langle W_n^{1/2} \rangle = \tilde{B}_1(k,\zeta_1,\zeta_2) \left(1 - (-1)^{-n\zeta_2}\right) \frac{\text{Ai}\left[C^{-1/3}\left(N - B - \frac{2n}{k(1-\zeta_2)}\right)\right]}{\text{Ai}\left[C^{-1/3}\left(N - B\right)\right]} + \\ + \tilde{B}_2(k,\zeta_1,\zeta_2) \left(1 - (-1)^{n\zeta_1}\right) \frac{\text{Ai}\left[C^{-1/3}\left(N - B - \frac{2n}{k(1+\zeta_1)}\right)\right]}{\text{Ai}\left[C^{-1/3}\left(N - B\right)\right]} \,. \tag{5.44}$$

An interesting observation regards the form of the vevs: as in the non-deformed case, the general form is a ratio of two Airy functions, whose coefficients and arguments are modified by the presence of masses. We can check our final results by expanding (5.42) and (5.44) for large values of $N$. In this limit, the two ratios lead to two different exponential behaviour, precisely as observed in (3.10), derived from the eigenvalue distribution at large $N$, and in (5.6), estimating the thermodynamic limit of the Fermi gas.

## 6 Strong coupling expansions in different regimes

In light of a physical interpretation of our results at strong coupling, we briefly recall some salient features of three-dimensional BPS Wilson loops from the AdS/CFT correspondence perspective. Subsequently, we will present the explicit expansions of the Fermi gas results obtained in the mass-deformed theory, discussing two relevant strong coupling limits.

### 6.1 Wilson loops and holography in the superconformal case

To begin with, we limit ourselves to the massless case, in which ABJM theory is dual to the M-theory on the background $\text{AdS}_4 \times S^7/\mathbb{Z}_k$. In the large $N$ limit, the common radius $L$ of $\text{AdS}_4$ and $S^7$ is a function of the gauge theory parameters

$$\left(\frac{L}{\ell_p}\right)^6 = 32\pi^2 kN \,, \tag{6.1}$$

where $\ell_p$ is the 11d Planck length. When $N$ is large and $k$ fixed, the M-theory is well approximated by the 11d SUGRA. In this regime, one can compute quantum corrections performing a perturbative $1/N$ expansion. We refer to them as the *M-theory limit* and *M-theory expansion*, respectively.

Another interesting limit of the correspondence can be identified as follows. We can think of $S^7$ as the Hopf fibration over $\mathbb{CP}^3$, where the fiber is the M-theory circle $S^1$ acted by the $\mathbb{Z}_k$. If $k$ is large enough, that is $k \gg N^{\frac{1}{5}}$ with $N \to \infty$, the M-theory circle shrinks to zero size and the dual configuration becomes the perturbative type IIA string theory on the background $\text{AdS}_4 \times \mathbb{CP}^3$. In the stringy regime, observables admit the genus expansion in the string coupling $g_{\text{s}} = \frac{2\pi i}{k}$

$$F(\lambda, g_{\text{s}}) = \sum_{g \geq 0} F_g(\lambda) \, g_{\text{s}}^{2g-2} \,, \tag{6.2}$$

where $\lambda \equiv \frac{N}{k}$ is the 't Hooft parameter of the gauge theory, and related to the string length $\ell_{\text{s}}$ via

$$\lambda = \frac{1}{32\pi^2} \left(\frac{L}{\ell_{\text{s}}}\right)^4 \,. \tag{6.3}$$

The genus expansion is nothing but the $1/N$ 't Hooft expansion in gauge theory as $g_{\text{s}} = \frac{2\pi i \lambda}{N}$. We shall refer to this regime as the *type IIA limit*.

Both the M-theory and 't Hoof expansion apply to the expectation values of BPS Wilson loops, allowing for precision tests of the AdS/CFT correspondence [46]. Indeed, if one limits to the maximally supersymmetric case, BPS Wilson loops in the fundamental representation are dual to a fundamental string ending along the path of the Wilson loop at the AdS boundary [59]. In the M-theory limit, the fundamental string is uplifted to an M2-brane extending through the M-theory circle [46]. One can also include the bosonic Wilson loops in this framework. The crucial point is that one has to smear the maximally symmetric solution for the fundamental string over a $\mathbb{CP}^1 \subset \mathbb{CP}^3$ to reproduce the reduced $SU(2) \times SU(2)$ R-symmetry [16, 61].

On the following, we shall compute the leading terms of the M-theory and the genus expansion for BPS Wilson loops in the presence of the mass deformations. We believe that similar holographic interpretations persist in the deformed theory and computational checks can be carried on also in this case. This is analogous, for instance, to the case of 4d $\mathcal{N} = 4$ SYM with a mass deformation [76, 77]. Gravitational backgrounds corresponding to ABJM with masses turned on were described and analyzed e.g. in [29, 51, 78]. However, we are not aware of any results for BPS Wilson loops in the dual theory. We are confident that our computations could provide a first result in that direction, calling for future comparisons on the holographic side.

## 6.2 The genus expansion

We would like to compute the 't Hooft expansion of the deformed vev of the $\frac{1}{2}$–BPS Wilson loop in the strong coupling limit $\lambda \gg 1$, with both the deformations turned on. This corresponds to the genus expansion in type IIA superstring theory. We make the substitution $k = \frac{N}{\lambda}$ and expand in powers of $1/N$, obtaining results that are valid in the strong 't Hooft coupling regime $\lambda \gg 1$.

The Wilson loop vevs have a genus expansion of the form

$$\langle W_n^{1/6,1/2} \rangle = \sum_{g=0}^{\infty} g_s^{2g-1} \langle W_n^{1/6,1/2} \rangle_g \,, \tag{6.4}$$

where $g_{\rm s}$ is related to the Chern-Simons coupling $g_{\rm s} = \frac{2\pi i}{k}$. We would like to compute the first term in eq. (6.4) for $\frac{1}{2}$–BPS Wilson loop, that corresponds to the genus zero or planar vev. Using the expansion for large arguments of the Airy function, recalling $\zeta \equiv \zeta_1 + \zeta_2$, and defining

$$
\begin{aligned}
\hat{B}_1(\zeta_1, \zeta_2) &= \left(1 - (-1)^{-n\zeta_2}\right) \frac{B_1(k, \zeta_1, \zeta_2)}{k} = \\
&= \left(1 - (-1)^{-n\zeta_2}\right) \frac{(1 - \zeta_2)}{4n\pi^2 (2 - \zeta) n!} \, i^{n(1-\zeta_2)} \, \Gamma\left(\frac{n\zeta}{2}\right) \Gamma\left(1 + n - \frac{n\zeta}{2}\right) \,,
\end{aligned}
\tag{6.5}
$$

we obtain

$$
\begin{aligned}
\langle W_\Box^{1/2} \rangle_{g=0,\zeta_{1,2}\neq 0} &= 2\pi i \, \hat{B}_1(\zeta_1, \zeta_2) \, e^{\frac{n\pi}{1-\zeta_2}\sqrt{2\lambda(1-\zeta_1^2)(1-\zeta_2^2)}} \left[ 1 - \frac{n\pi\sqrt{2(1-\zeta_1^2)(1-\zeta_2^2)}}{48(1-\zeta_2)\sqrt{\lambda}} + \right. \\
&\left. + \frac{n^2\pi^2(1-\zeta_1^2)(1+\zeta_2)}{2304(1-\zeta_2)\lambda} - \frac{n\pi\sqrt{(1-\zeta_1^2)(1-\zeta_2^2)}}{165888\sqrt{2}\,(1-\zeta_2)\lambda^{3/2}} \left( 72 + \frac{n^2\pi^2(1+\zeta_2)(1-\zeta_1^2)}{(1-\zeta_2)} \right) \right] + \\
&\quad + (\zeta_1 \leftrightarrow -\zeta_2) + \mathcal{O}(\lambda^{-2}) \,,
\end{aligned}
\tag{6.6}
$$

that is symmetric under the interchange $\zeta_1 \leftrightarrow -\zeta_2$ as expected. We observe the presence of two different exponential behaviour and of a non-trivial series in $1/\sqrt{\lambda}$ around them. We envisage that the exponents should correspond to different saddle-points of the fundamental string action suspended in the relevant gravitational background. From the one-loop

determinants around such solutions, the first term of the $1/\sqrt{\lambda}$ series could be hopefully recovered.

In the limit of $\zeta_1, \zeta_2 \to 0$ we find the expansion

$$\langle W_\Box^{1/2} \rangle_{g=0,\zeta_{1,2}=0} = 2\pi i^n \, e^{n\pi\sqrt{2\lambda}} \left( \frac{1}{4n\pi} - \frac{\sqrt{2}}{192\sqrt{\lambda}} + \frac{n\pi}{9216\lambda} - \frac{(n^2\pi^2+72)}{663552\sqrt{2}\,\lambda^{3/2}} \right), \qquad (6.7)$$

which is consistent with the results of [42], obtained in the undeformed case.

## 6.3 The M-theory limit beyond the leading order

The M-theory limit probes a different regime compared to the 't Hooft limit of the previous section. This limit is used to make contact with the eleven dimensional M-theory and is obtained keeping $k$ fixed while expanding in powers of $1/N$ (so $N \to \infty$).

The result with two deformations turns out to be

$$\langle W_\Box^{1/2} \rangle_{\zeta_{1,2}\neq 0} = \tilde{B}_1(k,\zeta_1,\zeta_2) \, \left( 1 - (-1)^{-n\zeta_2} \right) \, e^{\frac{n\pi}{1-\zeta_2}\sqrt{2\frac{N}{k}(1-\zeta_1^2)(1-\zeta_2^2)}} \left[ 1 - \frac{f(k,\zeta_1,\zeta_2)}{\sqrt{N}} + \right.$$

$$\left. + \frac{1}{2N} \left( \frac{n}{k(1-\zeta_2)} + f^2(k,\zeta_1,\zeta_2) \right) \right] + (\zeta_1 \leftrightarrow -\zeta_2) + \mathcal{O}(N^{-3/2}),$$

$$(6.8)$$

with

$$f(k,\zeta_1,\zeta_2) = \frac{\pi n \left[ \left(1-\zeta_1^2\right)\left(1-\zeta_2^2\right)k^2 + 4\left(\zeta_1^2+\zeta_2^2-6(\zeta_1^2-1)(\zeta_2+1)n+2\right)\right]}{24\sqrt{2}\,(1-\zeta_2)\sqrt{\left(1-\zeta_1^2\right)\left(1-\zeta_2^2\right)}\,k^{3/2}}. \qquad (6.9)$$

Also this expansion, as expected, is symmetric under $\zeta_1 \leftrightarrow -\zeta_2$. In the limit $\zeta_1, \zeta_2 \to 0$ we recover the known result:

$$\langle W_\Box^{1/2} \rangle_{\zeta_{1,2}=0} = \frac{i^{n-1}}{2} \csc\left( \frac{2n\pi}{k} \right) e^{n\pi\sqrt{\frac{2N}{k}}} \left( 1 - \frac{n\pi(k^2+24n+8)}{24\sqrt{2}\,k^{3/2}\,\sqrt{N}} + \right.$$

$$\left. + \left( \frac{n}{2k} + \frac{n^2\pi^2(k^2+24n+8)^2}{2304\,k^3} \right) \frac{1}{N} \right) + \mathcal{O}(N^{-3/2}), \qquad (6.10)$$

As explained in section 6 and proven recently in the undeformed case in [46], this limit allows to obtain information regarding the classical action and the one-loop fluctuations of a wrapped M2 brane in the dual $11d$ superstring theory with a mass-deformed AdS background. The holographic computation has not been performed to the best of our knowledge, and the calculation using the deformed metric involves some subtleties [29]. We limit ourself to observe that, also at the leading order in $N$, our result is modified by the masses in a non-trivial way. Both the exponential prefactors, including the argument of the $\frac{1}{\sin}$ coefficients, that still remains, are modified by $\zeta_1, \zeta_2$. The direct computation in M-theory would be an interesting application of AdS/CFT correspondence, generalizing in the deformed background the analysis of [46].

# 7 Wilson loops as conformal defects and integrated correlators

In this section, we present a second relevant application of our results, framing them in the context of defect conformal field theories. To begin with, we describe the Wilson loops from a slightly different corner, namely as a conformal defect.

The insertion of a BPS Wilson line in the massless ABJM model breaks the conformal group into the one-dimensional conformal group $SL(2, \mathbb{R})$, which fixes the line supporting the Wilson loop. All the relative details, including those on the superalgebra structure, can be found in [23]. Therefore, the BPS Wilson-loop described in Section 2 are examples of conformal defects [79]. The diminished symmetry combined with the possibility of defect operators -those that live on the defect- enlarges the set of observables. That is, the most general correlation function with the Wilson loop is of the form

$$\langle O_1(x_1) \ldots O_J(x_J)\hat{O}_1(t_1) \ldots \hat{O}_k(t_k)\rangle_W = \frac{\langle O_1(x_1) \ldots O_J(x_J)\hat{O}_1(t_1) \ldots \hat{O}_k(t_k)W\rangle}{\langle W\rangle} . \quad (7.1)$$

where the $O_i(x_i)$ are bulk operators, $\hat{O}_i(t_i)$ defect operators, with $t_i$ being the insertion point on the defect. The bracket $\langle\rangle_W$ indicates defect correlators with the Wilson loop normalized w.r.t. the vev of the Wilson loop [11] . The set of bulk and defect operators with their correlation functions constitutes the *defect conformal theory* of the Wilson loop.

Similar to ordinary CFTs, only a particular set of correlation functions is required to describe dCFTs. Apart from the standard bulk CFT information, dCFTs also need the defect spectrum, the OPE coefficients among defect operators, and a newly defined bulk-to-boundary OPE. The latter indicates that bulk operators in close proximity to the defect can be expressed in terms of defect operators. Namely, if we limit to scalar operators, for a bulk operator $O_J$ with dimension $\Delta_{O_J}$ and sitting at a distance $r$, we have schematically

$$O_J = \sum_k \frac{h_{J,k}}{r^{\Delta_{O_J} - \Delta_{\hat{O}_k}}} \hat{O}_k , \quad (7.2)$$

with $\Delta_{\hat{O}_k}$ being the dimension of the defect operator $\hat{O}_k$ and $h_{J,k}$ are understood as differential operators whose explicit form is fixed by the residual conformal symmetry [82]. When we plug (7.2) into a correlation function, we find that only the defect identity contributes. Therefore, we get

$$\langle O_J(x)\rangle_W = \frac{h_{O_J}}{r^{\Delta_{O_J}}} , \quad (7.3)$$

where $h_{O_J}$ are constants corresponding to the $h_{J,k}$ in (7.2). Indeed, non-vanishing one-point functions are one of the most distinctive features of dCFTs and contain useful physical dCFT data. While previous works on the defect theory in ABJM focused on correlation functions of defect operators [23, 81, 83], here we shall analyze bulk operators with defects[12].

---

[11]Because of such a normalization, we assume that defect correlation functions for the Wilson line can be computed equivalently using the Wilson line or the Wilson circle. Indeed, even if the vev of the circle operator is affected by a conformal anomaly [80], there is evidence that it does not affect defect correlators [81]. In the following, we will move from the Wilson loop to the Wilson line language without any further comment.

[12]See [84] for an alternative approach based on integrability.

In the following, we will relate the exact result for the Wilson loop in the presence of mass deformations to some of the coefficients $h_{O_J}$. The underlying idea is to treat the masses as small relevant perturbations around the defect CFT defined by the $\frac{1}{2}$–BPS Wilson loop in ABJM. Then, the mass derivatives of the mass-deformed vev provide information about the corresponding conformal defect theory. In the next section, we will make such an argument precise.

## 7.1 The strategy

The underlying method to establish the precise relation between defect CFT data and mass deformations in ABJM can be preliminarily introduced without the Wilson loop. Thus, we look for a relation between the mass-deformed partition function $Z[m]$ and bulk CFT data where, for future convenience, we set

$$\zeta_1 = 2imr \,, \qquad \zeta_2 = 0 \,. \tag{7.4}$$

In the language of Section 2 the corresponding deformation is a real mass deformation[13], realized at the linearized level by an operator $J$, namely

$$S[m] = S_{\text{ABJM}} + m \int d^3x \, J(x) + O(m^2) \,, \tag{7.5}$$

where $O(m^2)$ contains contact terms usually fixed by gauge invariance and supersymmetry.

To probe the undeformed ABJM model, we look at derivatives of the mass-deformed partition function and then set the mass to zero. At least at the linearized level, one has the relation

$$\frac{1}{Z} \frac{\partial^n}{\partial m^n} Z[m] \bigg|_{m=0} = \int d^3x_1 \ldots d^3x_n \, \langle J(x_1) \ldots J(x_n) \rangle. \tag{7.6}$$

This type of observables are called *integrated correlators* [12, 52, 86]. Of course, they can be defined for any deformations of a given CFT, but they are hard to handle, especially at strong coupling because of short distances singularities affecting the r.h.s. of (7.6). Moreover, additional terms from the non-linear part of $S[m]$ may contribute to (7.6).

If the supersymmetry is large enough [87], we can circumvent all these difficulties and extract physical information from (7.6). The way to see that, is to use the following cohomological Ward identity and write the mass dependent part of the action $S_{\text{mass}}$ as [55, 56]

$$S_{\text{mass}} \equiv \int_{S^3} \mathcal{L}_{\text{mass}} = \delta V - i4\pi r^2 m \oint_{S^1} d\tau \mathcal{J}(\tau) \,, \tag{7.7}$$

where $V$ is a functional whose explicit expression is not needed and $\mathcal{J}$ is the operator

$$\mathcal{J}(\tau) = U^I(\tau) \bar{V}_J(\tau) \operatorname{Tr}\big(C_I(\tau) \bar{C}^J(\tau)\big) \,,$$
$$U^I(\tau) = \frac{1}{\sqrt{2}} \left(e^{-\frac{i}{2}\tau}, \, 0 \,, \, e^{\frac{i}{2}\tau}, \, 0\right)^I \,, \quad \bar{V}_I(\tau) = \frac{1}{\sqrt{2}} \left(e^{\frac{i}{2}\tau}, \, 0 \,, \, -e^{-\frac{i}{2}\tau}, \, 0\right)_I \,, \tag{7.8}$$

---

[13]The analytic continuation in the deformation parameter of the results obtained with the Fermi gas is subtle. However, as argued in [85], it is well-defined for small masses. That is enough to perform mass derivatives.

where $\tau$ is the coordinate of a great circle on $S^3$. Crucially, the OPE of $\mathcal{J}(\tau)$ is non-singular [11]. Therefore, using (7.7), we can replace the 3d integrated correlators of (7.6) with the much more tractable integrated correlation functions of $\mathcal{J}$. Then, we get the exact relation

$$\left(\frac{i}{4\pi r^2}\right)^n \frac{1}{Z}\frac{\partial^n}{\partial m^n}Z[m]\bigg|_{m=0} = \oint_{S^1} d\tau_1 \cdots \oint_{S^1} d\tau_n \langle \mathcal{J}(\tau_1)\ldots\mathcal{J}(\tau_n)\rangle. \qquad (7.9)$$

As already discussed in Section 4, the l.h.s. of (7.9) can be computed with high precision, for example, with the Fermi gas. Then, (7.9) provides a powerful way to compute correlators and extract CFT data [12, 14, 52, 88].

Now, we are in business to apply the same type of technology, but with the addition of a BPS Wilson line [54, 89, 90]. The crucial point is that the Ward identity (7.7) depends only on the supercharge corresponding to $\delta$. Therefore, we can use it also in the presence of operators commuting with the supercharge. From that perspective, the configuration with $\frac{1}{2}$–BPS Wilson loop on a linked great circle with the one supporting the operators $\mathcal{J}(\tau)$ preserves the relevant supercharge and allows us to extend the machinery to defect correlation functions[14] [54].

In practice, we can start with the expectation value of the mass-deformed Wilson loop. Taking the mass derivatives lowers the complicated 3d integrated correlators, exactly as in (7.6), with the addition of the Wilson loop. The Ward identity of (7.7) reduces it to the 1d correlators in the presence of the Wilson loop. Then, we can write down the exact relation

$$\left(\frac{i}{4\pi r^2}\right)^n \frac{1}{W}\frac{\partial^n\langle W\rangle[m]}{\partial m^n}\bigg|_{m=0} = \oint_{S^1} d\tau_1 \cdots \oint_{S^1} d\tau_n \langle \mathcal{J}(\tau_1)\ldots\mathcal{J}(\tau_n)\rangle_W, \qquad (7.10)$$

where $\langle W\rangle[m]$ is the vev of the Wilson loop in ABJM, computed in the Section 4. Equation (7.10) provides a window to explore defect correlation functions in ABJM, even in the non-perturbative regime.

In the rest of the section, we apply the above ideas in two explicit examples, namely the first and second derivatives of the Wilson loop. In both cases, we will use (7.10) to extract the relevant defect CFT data.

## 7.2 The Bremsstrahlung

Let us begin with the simplest integrated correlator, namely the first derivative of $\langle W\rangle[m]$, which has a precise physical interpretation as the *Bremsstrahlung* function $B$ [54]

$$B = \frac{i}{8\pi^2 r}\frac{1}{\langle W_\square^{1/2}\rangle}\frac{\partial}{\partial m}\langle W_\square^{1/2}\rangle\bigg|_{m=0}. \qquad (7.11)$$

This quantity measures the energy emitted by the charged probe, whose worldline is described by the Wilson loop.

Equation (7.11) heavily relies on conformal invariance and supersymmetry. Using conformal invariance, we can interpret the Bremsstrahlung as the two-point function of the

---

[14]The method does not apply to the bosonic Wilson loop, as it is not annihilated by $\delta$.

displacement multiplet, namely the defect operator associated with the non-conservation of the components of the stress tensor orthogonal to the defect [91]

$$\partial_\mu T^{\mu i}(x) = -\delta_W(x^i)\mathsf{D}^i\,, \qquad \langle \mathsf{D}^i(t)\mathsf{D}^j(0)\rangle_W = \delta^{ij}\frac{12B}{|t|^{2\Delta_\mathsf{D}}}\,. \tag{7.12}$$

where the index $x^i$ indicates the components orthogonal to the defect, placed in $x^i = 0$, $\delta_W(x^i)$ is a delta function localized on the defect and $t$ is the coordinate along the defect. Moreover, because of the $\mathcal{N} = 6$ SUSY, the operator $\mathcal{J}$ is part of the stress tensor multiplet, thus implying that its one-point function is proportional to that of the stress tensor, which is a measure of the backreaction of the Wilson loop to a small deformation [21]. That intuition is made formal by a supersymmetric Ward identity establishing a precision relation among the one-point function of the stress tensor and the Bremsstrahlung [92], ultimately leading to our expression (7.11).

To evaluate (7.11), we consider the single-winding Wilson loop and perform the analytic continuation of (7.4). The explicit expression reads

$$\langle W\rangle(m) = -\frac{1}{2}\csc\left(\frac{2\pi}{k(1+2imr)}\right)\frac{\mathrm{Ai}\left[C^{-\frac{1}{3}}\left(N - B - \frac{2}{k(1+2imr)}\right)\right]}{\mathrm{Ai}\left[C^{-\frac{1}{3}}\left(N - B\right)\right]}\,. \tag{7.13}$$

From that, we obtain a compact formula for the Bremsstrahlung

$$B(k,N) = -\frac{1}{4\pi^2}\left[\left(\frac{2\pi}{k}\right)^{2/3}\frac{\mathrm{Ai}'\left[\left(\frac{2}{\pi^2 k}\right)^{-1/3}\left(N - \frac{k}{24} - \frac{7}{3k}\right)\right]}{\mathrm{Ai}\left[\left(\frac{2}{\pi^2 k}\right)^{-1/3}\left(N - \frac{k}{24} - \frac{7}{3k}\right)\right]} + \frac{2\pi}{k}\cot\left(\frac{2\pi}{k}\right)\right]\,. \tag{7.14}$$

The result agrees with all the previous derivations in the literature [24, 93–96]. That is another strong evidence of the correctness of our methods.

## 7.3 A new one-point function

Next, we exploit the formula (7.10), combined with the OPE and SUSY considerations, to extract a new defect one-point function. Even if we limit ourselves to consider the simplest example, namely the second derivative, in principle, one could look at higher derivatives and compute the one-point function of higher dimensional operators. From that perspective, our setup is the 3d analog of [97, 98].

We shall implement the OPE of $\mathcal{J}$ into defect correlators. That OPE is particularly tractable because of supersymmetry. Specifically, $\mathcal{J}$ is annihilated by a supercharge $Q_\beta$, provided that the operators are inserted into a great circle linked with the Wilson loop. That is the same kinematical configuration and the same supercharge such that (7.10) holds. Therefore, if we implement the OPE into a defect correlator, only operators in that cohomology of $Q_\beta$ contribute to the correlation function, significantly simplifying our task. In this way, instead of considering the full OPE of the stress-tensor multiplet [14, 52], we limit ourselves to its protected part.

The cohomology of $Q_\beta$ was fully characterized: the only operators contributing to the OPE are Lorentz scalars, whose dimensions are equal to a specific combination of the

R-charge [11]. Then, the most general OPE reads as[15]

$$\mathcal{J} \times \mathcal{J} = \lambda_0 \mathbb{1} + \lambda_2 \mathcal{X} \,, \tag{7.15}$$

where $\mathbb{1}$ is the identity operator, and $\mathcal{X}$ is an operator of dimension two, whose explicit form is not needed [16], and $\lambda_0$, $\lambda_2$ are OPE coefficients. The operator $\mathcal{X}$ is normalized as follows

$$\langle \mathcal{X} \mathcal{X} \rangle = \frac{1}{r^4} \,. \tag{7.16}$$

The OPE coefficients can be connected to the integrated correlators without the Wilson line of (7.9). Let us express some of the findings from [14, 52] in our notation. First of all, taking the vev of (7.15), we see that $\lambda_0$, which is also proportional to the central charge of the theory, is proportional to the second mass derivative

$$\lambda_0 = -\left(\frac{1}{8\pi^2 r^2}\right)^2 \frac{\partial_m^2 Z[m]}{Z} \,. \tag{7.17}$$

Subsequently, it is possible to connect $\lambda_2$ and the fourth mass derivative by iteratively applying the OPE (7.15) twice to the correlation function of four $\mathcal{J}$. This results in the exact formula

$$\lambda_2^2 = \left(\frac{1}{8\pi^2 r}\right)^4 \frac{\partial_m^4 Z[m]}{Z} - r^4 \lambda_0^2 \,, \tag{7.18}$$

which is equivalent, for example, to Eq. (3.10) of [14], up to a normalization.

Let us now discuss correlators with the Wilson loop. If we apply the bulk OPE in the presence of a defect, one-point functions are no longer vanishing. Then, we can relate these defect correlation functions to defect CFT data. Let us illustrate the case of the two-point functions $\langle \mathcal{J} \mathcal{J} \rangle_W$. With the Wilson loop, the only extra contribution is

$$\langle \mathcal{X} \rangle_W = \frac{h_\mathcal{X}}{r^2} \,. \tag{7.19}$$

Then, if we evaluate the OPE inside $\langle \mathcal{J} \mathcal{J} \rangle_W$, we find an exact relation for $h_\mathcal{X}$

$$h_\mathcal{X} = r^2 \frac{\langle \mathcal{J} \mathcal{J} \rangle_W - \lambda_0}{\lambda_2} \,. \tag{7.20}$$

Expressing all the CFT data in terms of mass derivatives and using (7.10), we find the explicit expression for $h_\mathcal{X}$[17]

$$h_\mathcal{X} = \frac{\frac{\partial_m^2 Z[m]}{Z} - \frac{\partial_m^2 W[m]}{W}}{\sqrt{\frac{\partial_m^4 Z[m]}{Z} - \left(\frac{\partial_m^2 Z[m]}{Z}\right)^2}} \,. \tag{7.21}$$

---

[15]In principle, $\mathcal{J}$ itself could appear in the OPE. However, it has been argued from symmetry and kinematical considerations that this is not the case [14, 52, 99]. That amounts to saying that the three-point function $\langle \mathcal{J} \mathcal{J} \mathcal{J} \rangle$ is zero, as it can be readily checked in ABJM using the formula (7.9).

[16]Presumably, $\mathcal{X}$ is a linear combination of a single and double trace operator built out of two couples $U^I C_I$ and $\bar{V}_I \bar{C}^I$, with $U^I$ and $\bar{V}_I$ defined in (7.8). The precise linear combination could be determined by solving the mixing problem between the single and double trace operators. A similar situation is described in more detail for the case $k = 1$ in [13].

[17]The expression for $h_\mathcal{X}$ suffers from a sign ambiguity coming from reading $\lambda_2$ from (7.18). One potential resolution is to look at the three-point function of $\langle \mathcal{J} \mathcal{J} \mathcal{X} \rangle$, but that goes beyond our goals. For the purposes here, it is assumed that $\lambda_2 > 0$.

In the following, we present explicit results for $h_\mathcal{X}$ at weak and strong coupling.

**Weak coupling** The first interesting limit is the weakly coupled limit $k \gg 1$. In this regime, the theory is weakly coupled and comparable with standard Feynman diagrams computations. These comparisons provide highly non-trivial tests for the whole formalism and can clarify subtle and interesting aspects, such as the framing anomaly in Chern-Simons matter theories [100]. The first non-vanishing term is of order $1/k^2$. Borrowing the results of [54, 88], we get

$$h_\mathcal{X} = \frac{\sqrt{2}\pi^2 N}{k^2 \sqrt{1+N^2}}. \tag{7.22}$$

**Strong coupling** For the strong coupling computations, we consider the derivatives of the Fermi gas result. As for the vev of the Wilson loop in Section 6, we analyze the M-theory and 't Hooft limit. We will briefly list the necessary expressions

$$Z = e^A C^{-\frac{1}{3}} \mathrm{Ai}\left[C^{-\frac{1}{3}}(N-B)\right], \tag{7.23}$$

with

$$C = \frac{2}{\pi^2 k (1+4r^2 m^2)}, \qquad B = \frac{\pi^2 C}{3} - \frac{1}{6k}\left(1 + \frac{1}{1+4r^2 m^2}\right) + \frac{k}{24}, \tag{7.24}$$

and

$$A(mr) = \frac{1}{4}\left(\mathcal{A}_{\mathrm{ABJM}}(k+2imr) + \mathcal{A}_{\mathrm{ABJM}}(k-2imr) + 2\mathcal{A}_{\mathrm{ABJM}}(k)\right). \tag{7.25}$$

$\mathcal{A}_{\mathrm{ABJM}}$ is the constant map function defined in (4.26). We find the following large $k$ expansion instrumental to expand derivatives in the 't Hooft limit [75]

$$\mathcal{A}_{\mathrm{ABJM}}(k) = 2\zeta'(-1) - \frac{1}{6}\log\frac{k}{4\pi} - \frac{\zeta(3)}{8\pi^2}k^2 + \sum_{g=2}^{\infty} \frac{4^g B_{2g} B_{2g-2}}{4g(2g-2)(2g-2)!}\left(\frac{2\pi i}{k}\right)^{2g-2}.$$

The expression for the Wilson loop with $n=1$ was already written in (7.13). It is now not so hard to study both the M-theory limit and the 't Hooft expansion. For the M-theory expansion we find

$$h_\mathcal{X} = -\frac{1}{\sqrt{2}} + \frac{3\pi\sqrt{\frac{1}{N}}}{k^{3/2}} + \frac{3\left(k - 4\pi\cot\left(\frac{2\pi}{k}\right)\right)}{\sqrt{2}k^2 N} \tag{7.26}$$

$$+ \frac{\left(\frac{1}{N}\right)^{3/2}\left(\left(18+\pi^2\right)k^2 + 48\pi\left(\pi\left(\cos\left(\frac{4\pi}{k}\right)+3\right)\csc^2\left(\frac{2\pi}{k}\right) - 2k\cot\left(\frac{2\pi}{k}\right)\right) - 160\pi^2\right)}{16\pi k^{5/2}}$$

$$+ \frac{k\left(18A''(0) + k^2 - 94\right) - 4\pi\left(k^2 - 52\right)\cot\left(\frac{2\pi}{k}\right)}{8\sqrt{2}k^3 N^2} + O(N^{-\frac{5}{2}}).$$

We find that the natural form of the 't Hooft expansion is

$$h_\mathcal{X} = \sum_{g\geq 0}\left(\frac{2\pi i}{k}\right)^{2g} h_\mathcal{X}^{(g)}. \tag{7.27}$$

with

$$h_\mathcal{X}^{(0)} = -\frac{1}{\sqrt{2}}\,, \tag{7.28}$$

$$h_\mathcal{X}^{(1)} = -\frac{3}{4\pi\sqrt{\lambda}} + \frac{3}{4\sqrt{2}\pi^2\lambda} - \frac{1}{64\pi\lambda^{3/2}} - \frac{9}{32\pi^3\lambda^{3/2}} + \frac{1}{32\sqrt{2}\pi^2\lambda^2}$$

$$- \frac{1}{2048\pi\lambda^{5/2}} - \frac{9}{512\pi^3\lambda^{5/2}} + \frac{1}{768\sqrt{2}\pi^2\lambda^3} + \mathcal{O}(\lambda^{-7/2})$$

Notice that even if the genus zero term does not depend on $\lambda$, this result does not extend to weak coupling. As $k$ becomes large, instantons are no longer negligible and presumably lead to the result of (7.22). However, it should be possible to recover that result from a string or M-theory computation.

## 8 Conclusions and future directions

In this paper we have extended the Fermi gas computations of BPS Wilson loops in the case of mass-deformed ABJM theories. We have presented the explicit 't Hooft and M-theory expansions that could be confronted with string and M-theory holographic calculations. From our expressions, we have derived the one-point and the two-point functions of a class of topological operators in the background of the 1/2–BPS Wilson loop, using the cohomological Ward identity of [54]. Emphasis has been given to the possible use of these results in the bootstrap program for the related DCFT and, as an example, we have computed a new one-point function of an operator of dimension two, appearing in the topological OPE. On the other hand, our computations could be considered just a starting point for a series of investigations, possibly going in different directions.

**Non-perturbative effects**   A natural continuation of this work would be to explore the non-perturbative contributions to the BPS Wilson loops from the Fermi gas perspective. Instanton corrections to the vacuum expectation value of 1/6–BPS Wilson loops in ABJM theory have been considered in [44][18], where it was found that the membrane instanton corrections are determined by the refined topological string in the Nekrasov-Shatashvili limit. The pole cancellation mechanism between membrane instantons and worldsheet instantons, originally discovered in [39] for the partition function, works also in the Wilson loop case. Actually, the non-perturbative structure of the mass-deformed partition function was already studied in [47] and the pole cancellation mechanism was claimed not to work in this case, calling for a better understanding of the instanton contributions. A different and maybe related open question is the possibility of a phase transition at large $N$, at a critical value of the deformation parameters [48, 49, 70, 71] for real valued masses. Moreover, [101] found evidence of a second-order phase transition for an infinite tower of values of the rank of the gauge group. The analysis presented in [85] suggests that non-perturbative contributions are no longer suppressed at the transition point and Wilson loops should sensibly change their behavior in the new regime.

---

[18]It was also noticed there some discrepancy with the results presented in [42]. The author kindly communicated to one of us (L.G.) similar disagreements with the computations performed in [24]. It would be nice to understand better the sources of these discrepancies, in light of our new calculations.

**Holography**   To the best of our knowledge, no holographic computation of BPS Wilson loops has been performed up to now in the deformed theory: our expansions provide precise expressions to be verified both for type IIA string and M-theory computations. Recent analog tests have been performed at M-theory level for the undeformed theory [46], while some aspects of the dual deformed holographic theory have been explored in [51]. It would also be interesting to explore the dual description of the correlation functions of chiral primaries in the presence of the Wilson line, confronting the results in the topological limit.

**Ward identities and OPE**   The study of ABJM theory in the presence of a 1/2–BPS Wilson line has mostly focused on the perspective of defect operators [23, 81, 83]. Instead, defect correlators with bulk operators are largely unexplored. As a primary step, it would be interesting to have a reformulation of all the 3d multiplets from the point of view of the BPS Wilson line. This is the starting point to derive Ward identities for the correlation function of bulk local operators, which are a crucial ingredient for engineering the superconformal bootstrap. The analog four-dimensional construction has been presented in [102, 103] and a possible strategy in three dimensions could be an extension of the formalism developed in [99]. One could then try to explore bulk-to-boundary OPE in this setting and use some constraints coming from the topological correlators. For instance, one can think of the Bremsstrahlung function as the OPE coefficient of the stress tensor with the *defect identity*. No bulk-to-boundary OPE is known for ABJM, even in the protected case. In the latter situation, one could get a non-perturbative universal condition as in [11]. For the non-protected case, a similar bootstrap problem can be found in [104].

**Integrated correlators**   We derived a result for $\langle W \rangle (m_1, m_2)$ with two masses. Expression like $\partial_{m_1} \partial_{m_2} \langle W \rangle (m_1, m_2)$ lead to integrated correlators of the form

$$\int d^3x \int d\tau \langle \mathcal{L}_{\text{mass}}(x) \mathcal{J}(\tau) \rangle_W \tag{8.1}$$

with $\mathcal{L}_{\text{mass}}(x)$ being the three-dimensional Lagrangian for the real mass deformations. Because the OPE of that is not necessarily limited to protected operators, these seem to be the most interesting objects to put severe constraints on CFT data.

That analysis may find remarkable applications in computing scattering amplitudes of gravitons by extended objects in string or M-theory [89]. Indeed, one should be able to express integrated correlators as Mellin amplitudes [105]. Then, provided that a flat space limit exists [106], one could send the AdS radius to infinity and recover the flat space amplitude. Using our result, it is likely that one can learn something about M2-branes or fundamental strings in type IIA beyond the SUGRA approximation.

## Acknowledgments

We thank Valentina Giangreco Marotta Puletti, Gabriel Bliard and Domenico Seminara for useful discussions and Carlo Meneghelli for clarifying remarks and important observations. This work has been supported in part by the Italian Ministero dell'Università e Ricerca

(MIUR), and by Istituto Nazionale di Fisica Nucleare (INFN) through the "Gauge and String Theory" (GAST) research project. The work of E.A. is supported by the Swiss National Science Foundation through the project 200020-197160 and through the National Centre of Competence in Research SwissMAP. This project has received funding from the European Research Council (ERC) under the European Union's Horizon 2020 research and innovation program (grant agreement number 949077). The work of L.G. has been supported by the OPUS grant no. 2022/47/B/ST2/03313 "Quantum geometry and BPS states" funded by the National Science Centre, Poland.

# Appendices

## A   Derivation of the generating functional of the Wigner-Kirkwood corrections $\mathcal{G}_r$

In this appendix, we provide technical details on the computation of the Wilson loop with the Fermi gas. In particular, we explain how to deal with the infinite corrections to the Hamiltonian. Those corrections are encoded in the generating functional of the Wigner-Kirkwood corrections $\mathcal{G}_r$, whose computation is our final goal

$$e_\star^{-tH_W} = (e^{-t\hat{H}})_W = \left( \sum_{r=0}^{\infty} \frac{(-t)^r}{r!} \mathcal{G}_r \right) e^{-tH_W} \,. \tag{A.1}$$

For our purposes, explained in Section 4, it is sufficient to work with a simplified form of the kinetic term

$$T(P) = aP = \frac{1}{2}(\mathrm{sgn}(P) - \zeta_2)P \,, \tag{A.2}$$

throughout the computation. The method is an extension of the procedure introduced in the massless case in [42]. Here, we review it, considering directly the presence of the masses.

To begin with, we consider a generating functional of the form

$$e_\star^{-tH_W} = e_\star^{-tG(Q)} \star e_\star^{-taP} \,, \tag{A.3}$$

Using the BCH formula, we write it as

$$e_\star^{-tH_W} = e_\star^{-tG(Q)} e^{\frac{i\hbar}{2} \overleftarrow{\partial}_Q \overrightarrow{\partial}_P} e_\star^{-taP} = \exp\left( -taP - te^{\frac{\xi t}{2}\partial} G(Q) \right) \,, \tag{A.4}$$

with $\partial = \partial_Q$, and $\xi = -ia\hbar$. We also used a property of the $\star$-product known as Bopp shifts

$$f_1(q,p) \star f_2(q,p) = f_1\left( q, p - \frac{i\hbar}{2}\overrightarrow{\partial}_Q \right) f_2\left( q, p + \frac{i\hbar}{2}\overleftarrow{\partial}_Q \right) \,, \tag{A.5}$$

and the fact that

$$e^{-\frac{i\hbar ta}{2}\overrightarrow{\partial}_Q} f(Q) = f\left( Q - \frac{i\hbar ta}{2} \right) \,. \tag{A.6}$$

To get the explicit form of $G(Q)$, we expand the star product and compare the result to the Hamiltonian in the form of (5.9). Applying the $\log_\star$ to the expression below

$$e_\star^{-tH_W} = e_\star^{-tG(Q)} \star e_\star^{-taP} = \exp_\star\left(-taP - t\sum_{m\geq 0} c_m(-ia\hbar t)^m G^{(m)}(Q)\right), \tag{A.7}$$

we derive, after some manipulations, the form of $G(Q)$ in terms of $U(Q)$

$$G(Q) = \frac{1}{t}\frac{1 - e^{-t\xi\partial}}{1 - e^{-\xi\partial}}U(Q). \tag{A.8}$$

At this stage, the generating function of all the $\mathcal{G}_r$ reads as

$$e_\star^{-tH_W} = \exp\left(-taP - \frac{e^{\frac{t\xi}{2}\partial} - e^{-\frac{t\xi}{2}\partial}}{1 - e^{-\xi\partial}}U(Q)\right). \tag{A.9}$$

Let us now rewrite it in a more convenient form for our upcoming computations. Using the formula (A.6), we recast the second term in the following way

$$\frac{e^{\frac{t\xi}{2}\partial} - e^{-\frac{t\xi}{2}\partial}}{1 - e^{-\xi\partial}}U(Q) = \sum_{l\geq 0}\frac{B_l(-1)^l}{l!}\xi^{l-1}\partial^{l-1}\left[U\left(Q + \frac{\xi t}{2}\right) - U\left(Q - \frac{\xi t}{2}\right)\right], \tag{A.10}$$

where the element $l = 0$ is taken to be

$$\frac{1}{\xi}\partial^{-1}\left[U\left(Q + \frac{\xi t}{2}\right) - U\left(Q - \frac{\xi t}{2}\right)\right] = t\sum_{g=0}^{\infty}\frac{1}{(2g+1)!}\left(\frac{t\xi}{2}\right)^{2g}U^{(2g)}(Q). \tag{A.11}$$

Applying the second term (A.10) to the expression for the potential in the semiclassical limit $|Q| \to \infty$

$$U(Q) = \frac{\zeta_1}{2}Q + \frac{|Q|}{2} + O(e^{-|Q|}), \tag{A.12}$$

we see that the only contribution comes from the terms $l = 0, 1$, as $U^{(n)}$ for $n \geq 2$ are exponentially suppressed when $|Q| \gg 1$. In particular, for $l = 0$ the only relevant term is for $g = 0$ since $U^{(2)}(Q) = \mathcal{O}(e^{-|Q|})$. Its expression is

$$\frac{1}{\xi}\partial^{-1}\left[U\left(Q + \frac{\xi t}{2}\right) - U\left(Q - \frac{\xi t}{2}\right)\right] = t\left(\frac{\zeta_1}{2}Q + \frac{|Q|}{2}\right). \tag{A.13}$$

For the $l = 1$ contribution, we should separate the cases of different sign of $Q$. In fact, it can be recast in

$$\frac{\xi t}{4}\left(\zeta_1 + \text{sgn}(Q)\right), \qquad \forall\, Q \neq 0. \tag{A.14}$$

Next, one should take into account corrections to the semiclassical limit. As explained in Section 4, we need the expression for the canonical density matrix for the following value of the parameter $t$

$$t = \frac{2n}{k(\text{sgn}(P) - \zeta_2)} \quad \Rightarrow \quad \frac{\xi t}{2} = -n\pi i. \tag{A.15}$$

With this value, it is easy to see that the deformations do not affect the large $Q$ behavior. Namely, we can write

$$U(Q) = \frac{|Q| + \zeta_1 Q}{2} + \tilde{U}(Q), \qquad \tilde{U}(Q) = \log(1 + e^{-Q}). \tag{A.16}$$

Therefore, we can safely extend the procedure of [42] to our case and argue that, for any $Q$, the first and second term of (A.10) are that of (A.13) and (A.14), respectively. We stress that the result holds only for the specific choice $t\xi = -2n\pi i$.

Then, since we can write

$$U\left(Q + \frac{\xi t}{2}\right) - U\left(Q - \frac{\xi t}{2}\right) = \frac{\xi t}{4}\left(\zeta_1 + \mathrm{sgn}(Q)\right), \tag{A.17}$$

the quantity resumming all the corrections is

$$\mathcal{S}(Q) = \frac{1}{1 - e^{-\xi\partial}}\left[\zeta_1 + \mathrm{sgn}(Q)\right] = \sum_{l\geq 0}\frac{B_l(-1)^l}{l!}\xi^{l-1}\partial^{l-1}\left[\zeta_1 + \mathrm{sgn}(Q)\right] = $$
$$= \frac{1}{\xi}\left(\zeta_1 Q + |Q|\right) + \frac{1}{2}\left(\zeta_1 + \mathrm{sgn}(Q)\right) + \mathcal{O}(\xi). \tag{A.18}$$

To calculate $\mathcal{S}(Q)$, we proceed in the same way as in [42]: we take the derivative w.r.t. $Q$ and multiply both sides by $1 - e^{-\xi\partial}$, so that

$$\mathcal{T}(Q) = \mathcal{S}'(Q) = \frac{\zeta_1}{\xi} + \frac{1}{1 - e^{-\xi\partial}}\left[\delta(Q) + \delta(-Q)\right]. \tag{A.19}$$

The second term was already computed in [42] and, borrowing their expression, we obtain

$$\mathcal{T}(Q) = \frac{i}{\theta}\coth\left(\frac{\pi Q}{\theta}\right) + \frac{\zeta_1}{\xi} = \frac{i}{\theta}\left(\coth\left(\frac{\pi Q}{\theta}\right) + \zeta_1\right), \tag{A.20}$$

where we set

$$\xi \equiv -i\theta = -ia\hbar = -i\pi k(\mathrm{sgn}(P) - \zeta_2). \tag{A.21}$$

Next, we integrate $\mathcal{T}(Q)$ w.r.t. $Q$ to obtain the expression for $\mathcal{S}(Q)$

$$\mathcal{S}(Q) = \frac{i}{\theta}\int dQ\left(\coth\left(\frac{\pi Q}{\theta}\right) + \zeta_1\right) = \frac{i}{\pi}\log\left(2\sinh\left(\frac{\pi Q}{\theta}\right)\right) + \frac{i\zeta_1}{\theta}Q + c, \tag{A.22}$$

where $c$ is the integration constant to be fixed in agreement with the expression (A.18) in the semiclassical limit. We guess the following expression for $\mathcal{S}(Q)$

$$\mathcal{S}(Q) = \frac{1}{2}\left(1 + \zeta_1\right) + \frac{i}{\pi}\log\left(2\sinh\left(\frac{\pi Q}{\theta}\right)\right) + \frac{i\zeta_1}{\theta}Q. \tag{A.23}$$

For $Q > 0$, this can be written as

$$\mathcal{S}(Q) = \frac{1}{2}\left(1 + \zeta_1\right) + \frac{Q}{\xi}\left(1 + \zeta_1\right) + \frac{i}{\pi}\log\left(1 - e^{-\frac{2\pi Q}{\theta}}\right), \tag{A.24}$$

while for $Q < 0$

$$\mathcal{S}(Q) = \frac{1}{2}\left(-1 + \zeta_1\right) + \frac{Q}{\xi}\left(-1 + \zeta_1\right) + \frac{i}{\pi}\log\left(1 - e^{\frac{2\pi Q}{\theta}}\right). \tag{A.25}$$

These two expressions can be combined $\forall\, Q \neq 0$ in

$$\mathcal{S}(Q) = \frac{1}{\xi}\left(|Q| + \zeta_1 Q\right) + \frac{1}{2}\left(\zeta_1 + \mathrm{sgn}(Q)\right) + \frac{i}{\pi}\log\left(1 - e^{-\frac{2\pi|Q|}{\theta}}\right). \tag{A.26}$$

For $Q \neq 0$, and $\xi$ small, we can expand the latter expression in

$$\mathcal{S}(Q) \approx \frac{1}{\xi}\left(|Q| + \zeta_1 Q\right) + \frac{1}{2}\left(\zeta_1 + \mathrm{sgn}(Q)\right), \tag{A.27}$$

consistently with (A.18).

We conclude that, for the simplified Hamiltonian

$$e_\star^{-H_W} = e_\star^{-U(Q)} \star e_\star^{-aP}, \tag{A.28}$$

the canonical density matrix with $t = \frac{2n}{k(\mathrm{sgn}(P)-\zeta_2)}$ is given by (in the general case of $P$ with either positive or negative sign)

$$\exp_\star\left(-\frac{2n}{k(\mathrm{sgn}\,(P) - \zeta_2)}H_W\right) = \exp\left[-ta|P| + n\pi i\mathcal{S}(Q)\right] =$$

$$= \exp\left[-\frac{n}{k}|P| + \frac{n\pi i}{2}\left(1 + \zeta_1\right) - \frac{n\zeta_1}{k(\mathrm{sgn}\,P - \zeta_2)}Q - n\,\log\left(2\sinh\frac{Q}{k(\mathrm{sgn}\,P - \zeta_2)}\right)\right]. \tag{A.29}$$

with the natural regularization procedure explained above. For $\zeta_1 = \zeta_2 = 0$ it recovers the result found in [42], as expected.

Now we should repeat the same calculations for an Hamiltonian with a simplified potential term

$$U(Q) = bQ, \quad \text{with} \quad b = \frac{1}{2}(\mathrm{sgn}(Q) + \zeta_1), \tag{A.30}$$

to derive the expression for the generating functional of the Wigner-Kirkwood corrections for region $B$. Following the same steps of the computation above, it is not hard to derive that

$$\exp_\star\left(-\frac{2n}{k(\mathrm{sgn}(Q) + \zeta_1)}H_W\right) =$$

$$= \exp\left[-\frac{n}{k}|Q| + \frac{n\pi i}{2}\left(1 - \zeta_2\right) + \frac{n\zeta_2}{k(\mathrm{sgn}\,Q + \zeta_1)}P - n\,\log\left(2\sinh\frac{P}{k(\mathrm{sgn}\,Q + \zeta_1)}\right)\right]. \tag{A.31}$$

# B  Explicit computation of $\lim_{\zeta_{1,2}\to 0}\langle W_n^{1/6}\rangle$

In this appendix, we shall perform the limit $\zeta_1, \zeta_2 \to 0$ of the vevs of BPS Wilson loop in eq. (4.1). The aim is to prove that the result coincides with the one directly derived in

[42] in the massless case. The limit is somewhat non-trivial due to the $1/\zeta$ singularity of the factor $\Gamma(n\zeta/2)$ for small values of $\zeta$. Because of this $1/\zeta$ behaviour, we must keep into account all the linear corrections in $\zeta_1$ and $\zeta_2$.

We compute the limit of the various factors of (4.1) one by one. Let us begin with the singular one. The terms with the Gamma functions have the following expansion

$$\Gamma\left(1 + n \mp \frac{n\zeta}{2}\right)\Gamma\left(\pm n\frac{\zeta}{2}\right) = \Gamma(n+1)\left[\pm\frac{2}{n\zeta} - H_n\right], \tag{B.1}$$

where $\gamma$ is the Euler gamma and $H_n$ are the harmonic numbers. Combining it with the following expansions

$$i^{n(1\pm\zeta_{1,2})} = i^n\left(1 \pm \frac{i\pi n\zeta_{1,2}}{2}\right),$$

$$\csc\left(\frac{2n\pi}{k(1\pm\zeta_{1,2})}\right) = \csc\left(\frac{2n\pi}{k} \mp \frac{2n\pi\zeta_{1,2}}{k(1\pm\zeta_{1,2})}\right) \approx \csc\left(\frac{2n\pi}{k}\right)\left(1 \pm \frac{2n\pi\zeta_{1,2}}{k}\cot\left(\frac{2n\pi}{k}\right)\right),$$

$$\frac{1}{2\pm\zeta} = \frac{1}{2} \mp \frac{\zeta}{4},$$
$$\tag{B.2}$$

we get the $\zeta \to 0$ limit of the coefficients $\tilde{B}_1, \tilde{B}_2$

$$\tilde{B}_1(k, \zeta_1, \zeta_2) = \frac{i^n}{4\pi}\csc\left(\frac{2n\pi}{k}\right)\left(\frac{2}{n\zeta} - H_n + \frac{1}{n} - \frac{i\pi\zeta_2}{\zeta} - \frac{4\pi\zeta_2}{k\zeta}\cot\left(\frac{2n\pi}{k}\right)\right) + \mathcal{O}(\zeta),$$

$$\tilde{B}_2(k, \zeta_1, \zeta_2) = \frac{i^n}{4\pi}\csc\left(\frac{2n\pi}{k}\right)\left(-\frac{2}{n\zeta} - H_n + \frac{1}{n} - \frac{i\pi\zeta_1}{\zeta} - \frac{4\pi\zeta_1}{k\zeta}\cot\left(\frac{2n\pi}{k}\right)\right) + \mathcal{O}(\zeta),$$
$$\tag{B.3}$$

where we did not include the $O(\zeta)$ terms because the limit of the Airy function does not contain additional singular terms.

Let us turn to those contributions. It is easy to see that the limit of the denominator is trivial up to $O(\zeta^2)$ corrections, that is

$$\text{Ai}\left[C^{-1/3}\left(N - \frac{k}{24} - \frac{2 + \zeta_1^2 + \zeta_2^2}{6k(1 - \zeta_1^2)(1 - \zeta_2^2)}\right)\right] = \text{Ai}\left[\tilde{C}^{-1/3}\left(N - \frac{k}{24} - \frac{1}{3k}\right)\right] + \mathcal{O}(\zeta^2), \tag{B.4}$$

with $\tilde{C} = \frac{2}{\pi^2 k}$. For the Airy functions in the numerator, we have to include also $\mathcal{O}(\zeta)$ terms. The expansions are

$$\text{Ai}\left[C^{-1/3}\left(N - \frac{k}{24} - \frac{(2 + \zeta_1^2 + \zeta_2^2)}{6k(1 - \zeta_1^2)(1 - \zeta_2^2)} - \frac{2n}{k(1 - \zeta_2)}\right)\right] =$$

$$= \text{Ai}\left[\tilde{C}^{-1/3}\left(N - \frac{k}{24} - \frac{6n + 1}{3k}\right)\right] - \frac{2n\zeta_2}{k}\tilde{C}^{-1/3}\,\text{Ai}'\left[\tilde{C}^{-1/3}\left(N - \frac{k}{24} - \frac{6n + 1}{3k}\right)\right] + \mathcal{O}(\zeta^2), \tag{B.5}$$

and, for symmetry arguments,

$$\text{Ai}\left[C^{-1/3}\left(N - \frac{k}{24} - \frac{(2 + \zeta_1^2 + \zeta_2^2)}{6k(1 - \zeta_1^2)(1 - \zeta_2^2)} - \frac{2n}{k(1 + \zeta_1)}\right)\right] =$$

$$= \text{Ai}\left[\tilde{C}^{-1/3}\left(N - \frac{k}{24} - \frac{6n + 1}{3k}\right)\right] + \frac{2n\zeta_1}{k}\tilde{C}^{-1/3}\,\text{Ai}'\left[\tilde{C}^{-1/3}\left(N - \frac{k}{24} - \frac{6n + 1}{3k}\right)\right] + \mathcal{O}(\zeta^2). \tag{B.6}$$

Putting all together, we see that the divergent terms of $\mathcal{O}(\zeta^{-1})$ cancel in a non-trivial way and we are left with

$$
\langle W_n^{1/6} \rangle = -\tilde{C}^{-1/3} A_1(k) \frac{\text{Ai}'\left[\tilde{C}^{-1/3}\left(N - \frac{k}{24} - \frac{6n+1}{3k}\right)\right]}{\text{Ai}\left[\tilde{C}^{-1/3}\left(N - \frac{k}{24} - \frac{1}{3k}\right)\right]}
$$
$$
+ A_2(k) \frac{\text{Ai}\left[\tilde{C}^{-1/3}\left(N - \frac{k}{24} - \frac{6n+1}{3k}\right)\right]}{\text{Ai}\left[\tilde{C}^{-1/3}\left(N - \frac{k}{24} - \frac{1}{3k}\right)\right]} + \mathcal{O}(\zeta) ,
$$
(B.7)

where

$$
A_1(k) = \frac{2\pi n}{k} \csc\left(\frac{2\pi n}{k}\right) \beta_1(k) ,
$$
$$
A_2(k) = \frac{2\pi n}{k} \csc\left(\frac{2\pi n}{k}\right)\left[\left(\frac{k}{2n} - \pi \cot\left(\frac{2\pi n}{k}\right)\right)\beta_1(k) + \beta_2(k)\right] ,
$$
(B.8)
$$
\beta_1(k) = \frac{i^n}{2\pi^2 n} \quad , \quad \beta_2(k) = -\frac{k}{4\pi^2 n} i^{n+1}\left(\frac{\pi}{2} - iH_n\right) .
$$

That is the same result found in the ABJM limit in [42].

A similar analysis can be carried on for the 1/2–BPS Wilson loop. Following the same steps, one finds again that the limit $\zeta \to 0$ gives back the result of [42].

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
