# Peer review of "BPS Wilson loops in mass-deformed ABJM theory: Fermi gas expansions and new defect CFT data"

_SciPost Physics_

## Round 3 · Referee Report · Anonymous (Referee 1) · 2024-3-25

Report
Requested changes
-
on top of page 9, "function" should be "functions"
-
page 9, "is found" should be "is found to be"
-
page 10, "we remark the" should be "we note the"
-
page 11 "which might result technical" should be "which might seem technical"
-
could you clarify the meaning of terms like (-1)^{n zeta} ? are these complex for general mass?
-
page 28, the mass deformation in 7.5 should also include the Delta=2 operator denoted as K in 3.4 of 1808.10554. After going to 1d, this operator drops out, but in 3d it appears, and would be important if one considered mixed mass derivatives where you cant go to the 1d theory.
We thank the referee for the helpful comments, which contributed to raising the quality and clarity of the paper. We reply to the comments below.
For point 5: As the referee pointed out, the terms $(-1)^{n\zeta}$ are in general complex, with the convention explained in footnote 10. Therefore, the vev of the 1/6-BPS loop is generically complex. That is not surprising as it is not a real operator, an aspect that does not depend on the presence of the masses, as one can see, e.g., from eq 4.110-4.113 of 1207.0611. The vev of the 1/2-BPS operator in the fundamental representation is either real or imaginary, depending on the winding number n. The referee can see that from 2.13, and using that the vev of the Wilson loop for the second U(N) factor, namely $\langle\hat{W}\rangle$, is the complex conjugate of the vev of the Wilson loop for the first U(N), that is $\langle W\rangle$. Then, if n is odd, the result will be a real number, otherwise an imaginary one. Indeed, one can combine the various $(-1)^{n\zeta}$ and show this explicitly. We will write the explicit expression for the 1/2-BPS Wilson loop in a revised version. Accordingly, we will modify the definition of the coefficients $\tilde{B}_1$ and $\tilde{B}_2$ as needed in the paper.
For point 6: We will clarify the notations in equation 7.5 following the referee's observation, specifying that J is a dimension one operator and K is a dimension two operator. We will also make the notation of 7.6 consistent with the new notation.
For points 1 to 5: We will implement all the other corrections.

Author: Luigi Guerrini on 2024-05-06 [id 4474]
(in reply to Report 4 on 2024-04-20)We thank the referee for the comments and suggestions. We will implement all the requested changes in a revised version.

---

## Round 3 · Referee Report · Anonymous (Referee 2) · 2024-4-11

Strengths
1- The paper computes new exact observables in certain supersymmetric 3d theories. 2- The paper finds connections between the calculated observables and previously studied observables.
Weaknesses
1- The writing is poor. It shifts back and forth from attempting to be pedagogical to very specialized jargon.
Report
The paper also calculates correlation functions of local operators and BPS Wilson loops, especially something known as integrated correlators and finds connections with defect-CFT data. These again are well established techniques applied in this new setting with new, correct and interesting results.
My main issue with the paper is the style of writing, where it is unclear who the intended audience of the paper is. The presentation oscillates from attempts at pedagogical introduction to different tools to very specialize technical colloquialisms. An example of the first is the review of the fermi gas approach and even the explanation of basic facts that matrix models can be solved by studying eigenvalue distributions around eqn (3.1). As an example of the second is the term mass-deformed ABJM, used throughout and only briefly explained around (2.1), but not in a way that can be understood by someone who hasn't studied ABJM theory previously. Another example is "maximally supersymmetric case" in the abstract. Does that refer to k=1,2? To no mass deformation? Still not clear to me.
I could give a long list of examples of this, but at the end it makes the paper very hard to read. As an expert, I can just skip the reviews, but I still get confused by the private jargon. I don't expect anybody who is not an expert to be able to follow the various reviews, so the paper would be much better if it served the intended audience of experts and relied on a consistent language and set of tools.
Another illustration of that is that the introduction includes many references, but then on page 4 there are only 2 general references. Is page 4 really a self-contained review? It would be far better to remove most of it and rely on a good selection of references. The introduction also has no formulas, but relies on specialized words to try to convey technical details and it simply doesn't work. It goes through theories and observables that were studied and those that are new to this paper when a few key formulas would convey the message far better than the linguistic gymnastics attempting to convey technical details in regular speech.
Requested changes
1- Rewrite the introduction to include the main ingredients studied in paper. It would be good if (2.3) and (2.14) are already on the first page, where one can then discuss the parameters zeta. This will make it far clearer what is studied in this paper rather than talking about Wilson loops in mass deformed ABJM. Same for the topological sector and integrated correlators.
2- Remove section 2. Parts of it should go in the introduction.
3- Remove the introduction to sec. 3.
4- Remove sec. 4. Include the minimal introduction and reference at the start of section 5.
5- Shorten the introduction to sec. 7.
6- add ref to 2304.01924 (the reviewer is not an author of that paper).
7- Top of page 3: what is "holographic level"?
8- spelling of 't Hooft
9- Clarify what is meant by topological operators have dimension 1. Topological operators are dimensionless...
10- Change comment above (3.11). There is nothing novel about getting two exponents, it just means that you include two saddle points. Normally one only keeps leading saddle points, but when using localization one actually keeps all saddle points. In this case, this is at the level of the matrix model and indicates that the saddle point analysis is a bit ambiguous, but there is nothing striking about that.
Recommendation
Ask for minor revision
We thank the referee for the detailed comments.
Before responding to the points raised by the referee, we would like to address the general comment on our writing. Although our investigation may seem slightly technical and somewhat specific at first glance, we believe the final results may interest readers with diverse backgrounds. For instance, possible future directions intersect with SUSY QFT, bootstrap, and holography. However, methods like the Fermi gas, for example, may be unfamiliar to those without prior exposure to the topic. Therefore, we have chosen to include review sections and strive to present the logic and results of our article without assuming any prior research background of the reader.
Despite the referee's opinion, we still believe that the target audience for our paper should not be limited to specialists. Moreover, the reports of the other two reviewers, who find the paper clear as is, suggest that no stylistic changes are necessary. However, some of the referee's observations are indeed valid, can enhance the quality of our work, and do not conflict with our objectives. Therefore, we propose the following changes to enhance the effectiveness of our introductory sections and minimize the use of jargon.
1-We will merge the introduction with Section 2 into a single introduction. We will organize it into a non-technical overview, highlighting the formulas suggested by the referee. We will explain their physical origins in a subsection of the introduction, while also clarifying any necessary technical terms using formulas.
2- We will shorten the introduction to Sec 3 and Sec 7.
3-Concerning section 4, we retain it because it introduces the concepts and formulas needed for Sec 5. Moreover, this type of introduction is characteristic of many papers on the Fermi gas (e.g., 2004.13603 ). Finally, as specified below eq 4.3, our presentation is aimed at non-experts, who can find the essential ideas behind the computation and some more manageable related applications.
4- We cite the reference 2304.01924 when we discuss the bremsstrahlung function in sec 7.2. Does the referee want the citation in another specific part of the paper?
5- By "holographic level" we mean the framework of a mass-deformed version of AdS_4/CFT_3 of 1302.7310 or 2304.12340. We will modify the ambiguous sentence.
6-For the sentence in the abstract, we mean the 1/2-BPS Wilson loop, but the expression is confusing. We will fix the ambiguity.
7-Topological operators are indeed dimensionless. However, they come from a twist of physical operators with a given scaling dimension. That is what we meant by the dimension of topological operators. We will clarify the terminology in the revised version.
8- We will fix the typos and remove the word novel.

---

## Round 3 · Referee Report · Anonymous (Referee 3) · 2024-4-20

Strengths
1) The paper contains new original results for 1/6 and 1/2 BPS Wilson loops in the two-mass deformed ABJM theory 2) The calculations are solid and consistent with the zero-mass limit 3) The Wilson loop results can be the starting point for developing an interesting example of non-conformal holography 4) New results for one and two- point functions on the Wilson loop defects are interesting for further developing the bootstrap program on the defect
Weaknesses
Report
Requested changes
1) Page 6 – last paragraph below eq. (2.10): the “weaker condition” was originally proposed in ref [18] and later exploited in [62]. It should be fair to refer also to [18] there.
2) Second line after eq. (4.18): “an useful” should be “a useful”
3) Footnote 16, at page 31: in the first line, “operator” should be “operators”.
4) First line of section 8: maybe with “extended” the authors meant “generalized”. This should be corrected.
Recommendation
Ask for minor revision

---

## Editorial Decision

unknown